# Masked Frequency Modeling for Self-Supervised Visual Pre-Training

**Jiahao Xie**[1,2], **Wei Li**[1,2], **Xiaohang Zhan**[3], **Ziwei Liu**[1,2], **Yew Soon Ong**[2,4], **Chen Change Loy**[1,2]
[1]S-Lab, NTU    [2]SCSE, NTU    [3]CUHK    [4]A*STAR, Singapore
{jiahao003, wei.l, ziwei.liu, asysong, ccloy}@ntu.edu.sg
xiaohangzhan@outlook.com

## Abstract

We present **M**asked **F**requency **M**odeling (**MFM**), a unified frequency-domain-based approach for self-supervised pre-training of visual models. Instead of randomly inserting mask tokens to the input embeddings in the spatial domain, in this paper, we shift the perspective to the frequency domain. Specifically, MFM first masks out a portion of frequency components of the input image and then predicts the missing frequencies on the frequency spectrum. Our key insight is that predicting masked components in the frequency domain is more ideal to reveal underlying image patterns rather than predicting masked patches in the spatial domain, due to the heavy spatial redundancy. Our findings suggest that with the right configuration of *mask-and-predict* strategy, both the structural information within high-frequency components and the low-level statistics among low-frequency counterparts are useful in learning good representations. For the first time, MFM demonstrates that, for both ViT and CNN, a simple non-Siamese framework can learn meaningful representations even using **none** of the following: (i) extra data, (ii) extra model, (iii) mask token. Experimental results on image classification and semantic segmentation, as well as several robustness benchmarks show the competitive performance and advanced robustness of MFM compared with recent masked image modeling approaches. Furthermore, we also comprehensively investigate the effectiveness of classical image restoration tasks for representation learning from a unified frequency perspective and reveal their intriguing relations with our MFM approach. Project page: https://www.mmlab-ntu.com/project/mfm/index.html.

## 1 Introduction

Following the success of Masked Language Modeling (MLM) such as BERT (Devlin et al., 2019) in natural language processing (NLP), Masked Image Modeling (MIM) (Bao et al., 2022; He et al., 2022; Wei et al., 2022; Xie et al., 2022) has shown promising performance in self-supervised pre-training of visual models. Both MLM and MIM follow a common *corrupt-and-predict* paradigm – randomly masking a portion of input data and then learning to predict the missing parts. This simple recipe enables modern Transformer-based deep architectures (Vaswani et al., 2017; Dosovitskiy et al., 2020) to learn generalizable representations from ubiquitous unlabeled text or image data.

By default, current MIM methods such as BEiT (Bao et al., 2022), MAE (He et al., 2022) and SimMIM (Xie et al., 2022) perform masking in the spatial domain by excluding image patches randomly, a strategy inspired by MLM that performs masking on words (Figure 1(a-b)). However, unlike human-generated language that is succinct and highly semantic, raw pixel values in the spatial domain are of low information density. To cope with heavy spatial redundancy in images, MAE (He et al., 2022) shows that one would need to mask a very high proportion (*e.g.*, 75%) to encourage the learning of meaningful features.

Beyond masking image patches, which is a particular way of corruption, in this paper, we are interested in investigating the effectiveness of other corruption strategies for self-supervised representation learning. We first explore the corruption recipes commonly applied in low-level image processing tasks, including image super-resolution (SR), deblurring and denoising. As shown in

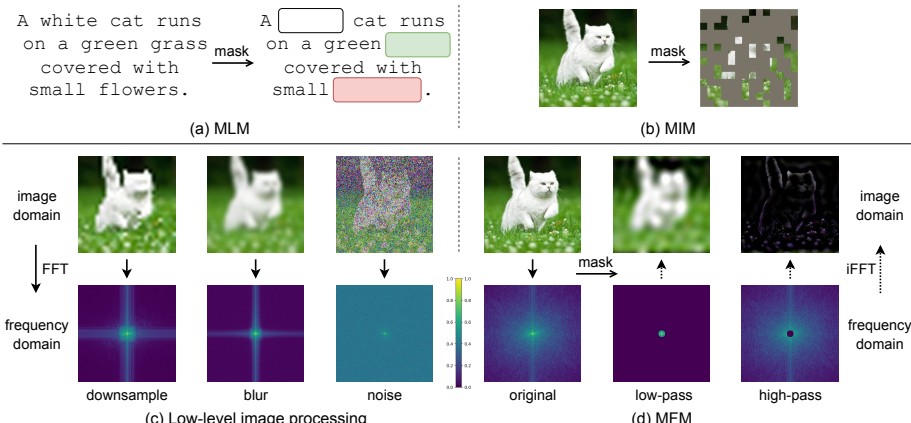

Figure 1: **Comparison of masking recipes** in Masked Language Modeling (MLM), Masked Image Modeling (MIM), low-level image processing and Masked Frequency Modeling (MFM). Note the differences of masked information among MIM, low-level image processing and MFM.

Figure 1(c), the downsampling, blur, and noise operations can degrade the exemplar image effectively in the spatial domain, thus potentially serving as useful corruption strategies. However, the corruption induced in the spatial domain prevents us from analyzing what specific information is corrupted and needs to be reconstructed. To better understand these low-level corruptions, we shift our attention from the spatial image domain to the frequency domain.

In the frequency domain, one could observe underlying patterns of an image not conveniently visible from raw pixel values. For example, the downsampling and blur operations dominantly remove the high-frequency image details, while adding noises tends to corrupt the full frequency spectrum of an image globally (Figure 1(c)).

Driven by this observation, we present a simple and effective masking strategy in the frequency domain for self-supervised visual representation learning, dubbed as **M**asked **F**requency **M**odeling (**MFM**). Specifically, we first perform Fast Fourier Transform (FFT) to convert each input image into its frequency representation, *i.e.*, frequency spectrum. We then mask a portion of frequencies on the frequency spectrum using a low-/high-pass filter. With inverse FFT (iFFT), we finally take the corrupted image with some of the frequencies attenuated as input. Our encoder is quite flexible as no mask tokens are inserted. Thus, MFM can embrace both the vision Transformer (ViT) (Dosovitskiy et al., 2020) and convolutional neural network (CNN) (LeCun et al., 1989) families. Our decoder is a lightweight linear layer that reconstructs the masked frequency values on the frequency spectrum via a frequency loss. As shown in Figure 1(d), an image with low or high frequencies attenuated would reveal entirely different patterns: the low-frequency components usually contain object smooth structure such as colors and styles, while the high-frequency counterparts largely depict the object outline or silhouette structure. Such unique properties of the frequency domain make it appealing for reducing information redundancy, thus creating a nontrivial and meaningful self-supervisory task.

Our contributions are summarized as follows:

**1)** We propose a new masked frequency modeling task to pre-train visual encoders in a self-supervised manner. Our MFM is agnostic to the architectures, and we demonstrate the flexibility of applying MFM for both ViT and CNN families.

**2)** We contribute the first study of low-level corruption tasks for self-supervised learning (SSL) in frequency domain. We investigate the effectiveness of corruption strategies commonly adopted in low-level image processing tasks (*i.e.*, SR, deblurring and denoising) for SSL from a unified frequency perspective and reveal that the representation learning capability of these corruption tasks actually depends on the architectures: they can achieve comparable and even better results than their supervised counterpart on ViT, but no gains are observed on CNN.

**3)** Extensive experiments show that our MFM can achieve competitive performance among existing MIM approaches on downstream tasks, such as image classification and semantic segmentation, while not using mask tokens or other more complex designs. Further analysis on several robustness benchmarks also exhibits more appealing robustness of the studied corruption tasks than MIM.

## 2    RELATED WORK

**Masked language modeling** and its auto-regressive variants, such as BERT (Devlin et al., 2019) and GPT (Radford et al., 2018; 2019; Brown et al., 2020), have achieved great success in pre-training large-scale language models in the NLP community. These approaches perform masking on the human-generated language by holding out random words and then predicting the missing content. This simple *mask-word* recipe has shown excellent ability in pre-training generalizable representations for broad NLP applications.

**Masked image modeling** leverages images corrupted by masking to learn useful representations. Pioneered with stacked autoencoders (Vincent et al., 2010) and context encoders (Pathak et al., 2016) using CNNs, recent approaches (Bao et al., 2022; He et al., 2022; Xie et al., 2022; Wei et al., 2022; Chen et al., 2022) follow the *mask-word* strategy in NLP to randomly mask image patches in the spatial domain using the vision Transformers (Dosovitskiy et al., 2020; Liu et al., 2021). Along with this *mask-patch* strategy, different types of prediction targets have been studied, including discrete tokens (Bao et al., 2022; Dong et al., 2021), raw pixels (He et al., 2022; Xie et al., 2022), and hand-crafted features (Wei et al., 2022). Besides, iGPT (Chen et al., 2020a) takes a low-resolution image sequence as input and predicts missing pixels in an auto-regressive manner. Several methods (Zhou et al., 2022; El-Nouby et al., 2021) also integrate MIM into contrastive-based Siamese frameworks. Our work differs from previous approaches in that we perform masking in the frequency domain, which relies on *none* of the following: (i) extra data (Bao et al., 2022; Dong et al., 2021; Fang et al., 2022), (ii) extra model (Zhou et al., 2022; El-Nouby et al., 2021; Fang et al., 2022; Shi et al., 2022; Chen et al., 2022), or (iii) mask token (Bao et al., 2022; He et al., 2022; Xie et al., 2022; Wei et al., 2022; Chen et al., 2022). CIM (Fang et al., 2022) also does not use mask token. However, introducing an auxiliary generator to corrupt the input images adds nontrivial pre-training overhead. In contrast, our frequency-domain-based corruption strategy can achieve comparable performance with negligible computational cost.

**Self-supervised learning** mainly focuses on designing effective pretext tasks for pre-training (Doersch et al., 2015; Wang & Gupta, 2015; Noroozi & Favaro, 2016; Larsson et al., 2016; Zhang et al., 2016; 2017c; Noroozi et al., 2017; Bojanowski & Joulin, 2017; Pathak et al., 2017; Gidaris et al., 2018). Contrastive learning (Wu et al., 2018; He et al., 2020; Misra & Maaten, 2020; Chen et al., 2020b;c; Grill et al., 2020; Chen & He, 2021; Chen et al., 2021; Caron et al., 2021) has dominated the field over the past few years. Unlike the *mask-and-predict* pretext task, contrastive learning typically uses a Siamese framework and greatly relies on data augmentation.

**Low-level image processing** tasks, such as image super-resolution (Dong et al., 2015), deblurring (Zhang et al., 2022) and denoising (Zhang et al., 2017b), focus on restoring the high-fidelity image from its corrupted input. The corrupted images are usually generated with degradation transformations, which consist of downsampling, blur, noise and JPEG compression. Recent promising results of MIM motivate us to investigate the effectiveness of these corruption operations in the context of representation learning.

**Frequency domain analysis** has been widely adopted in many computer vision tasks, such as image generation (Jiang et al., 2021), domain adaptation (Xu et al., 2021), and image super-resolution (Pang et al., 2020). Early studies (Oppenheim et al., 1979; Oppenheim & Lim, 1981; Piotrowski & Campbell, 1982; Hansen & Hess, 2007) have revealed that in the frequency domain, the phase component largely captures high-level semantics of the original signals, while the amplitude component mainly retains low-level statistics. As such, underlying image patterns can be more conveniently observed in the frequency representation, compared with the raw pixel values in the spatial domain. Motivated by the intriguing properties of the Fourier domain, we propose a novel *mask-frequency* recipe and conduct *the first study w.r.t.* masked information modeling in the frequency domain for image data.

## 3    APPROACH

Our masked frequency modeling (MFM) is a simple yet effective self-supervised pre-training approach, which masks out a portion of image frequency components and predicts the missing frequencies on the frequency spectrum. Figure 2 shows the overview of our approach. The framework consists of four components: masking strategy, encoder, decoder, and reconstruction target. We first

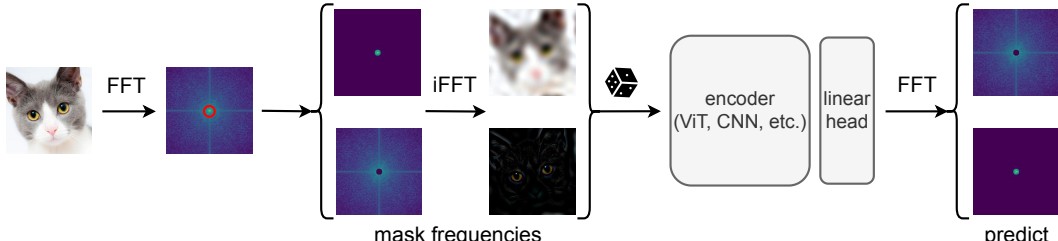

Figure 2: **Overview of our MFM pre-training pipeline.** We convert each input image into frequency domain via FFT and mask a portion of frequencies on the frequency spectrum via a low-pass (*top*) or high-pass (*bottom*) filter. After iFFT, the low-/high-pass filtered spatial images are then randomly fed to the encoder (*e.g.*, ViT, CNN), with a lightweight one-layer head to predict the masked frequency values on the frequency spectrum via a frequency loss. The red circle denotes the selected mask radius, and the dice icon refers to the random sampling process of low-/high-pass filters, following a Bernoulli distribution.

detail each component of MFM in Section 3.1, and then discuss the relation of our approach with low-level image processing tasks in Section 3.2.

## 3.1 MASKED FREQUENCY MODELING

**Preliminary: Frequency representation of images.** Given a single channel image[1] $x \in \mathbb{R}^{H \times W}$, we can obtain the corresponding frequency representation via 2D Discrete Fourier Transform $\mathcal{F}(x)$:

$$\mathcal{F}(x)(u,v) = \sum_{h=0}^{H-1} \sum_{w=0}^{W-1} x(h,w) e^{-i2\pi \left( \frac{uh}{H} + \frac{vw}{W} \right)}, \tag{1}$$

where $x(h,w)$ is the real pixel value at the coordinate of $(h,w)$ on the spatial image, $\mathcal{F}(x)(u,v)$ is the complex frequency value at the coordinate of $(u,v)$ on the frequency spectrum, $e$ and $i$ are Euler's number and the imaginary unit, respectively. Accordingly, $\mathcal{F}^{-1}(x)$ defines the inverse Fourier transform that maps spectral signals back into original image space. Both the Fourier transform and its inverse can be calculated efficiently using the FFT algorithm (Nussbaumer, 1981).

**Masking strategy.** We define a mask $M \in \{0,1\}^{H \times W}$, whose value is determined by a thresholding function that separates the low and high frequency components from $\mathcal{F}(x)$ according to a hyper-parameter, *i.e.*, radius $r$:

$$M(u,v) = \begin{cases} 1, & \text{if } d((u,v),(c_h,c_w)) < r \\ 0, & \text{otherwise} \end{cases} \tag{2}$$

where $(c_h, c_w)$ denotes the center of the image, $d(\cdot, \cdot)$ denotes a certain distance criterion. Here, we use the Euclidean distance, *i.e.*, a circle mask as default. Note that the mask shape is not solely restricted to a circle one, and we study the effects of different mask shapes in the experiment section.

With the predefined mask $M$, we can easily obtain the decomposed low-pass filtered image $x_l$ and the high-pass filtered counterpart $x_h$ as follows:

$$x_l = \mathcal{F}^{-1}(\mathcal{F}(x) \odot M), \quad x_h = \mathcal{F}^{-1}(\mathcal{F}(x) \odot (\mathbb{1} - M)), \tag{3}$$

where $\mathbb{1}$ is the all-ones matrix, $\odot$ is the Hadamard product between matrices. These filtered images are then randomly selected with a Bernoulli distribution and fed to an encoder as input[2].

**MFM encoder.** The architecture of our encoder is quite flexible since we do not insert any mask tokens on the corrupted non-overlapping patch embeddings as in MIM (Bao et al., 2022; He et al., 2022; Xie et al., 2022; Wei et al., 2022). Therefore, our MFM can be applied on both ViT and CNN architectures without any special designs. In this paper, we mainly use a standard ViT (Dosovitskiy et al., 2020) as our encoder for a direct comparison with MIM methods. Specifically, we first divide a

---

[1]For RGB images, the procedure is operated on each channel independently.
[2]Despite performing masking in the frequency domain, we still take the converted spatial images as input such that our model would not suffer an input domain gap between pre-training and fine-tuning.

filtered spatial image into regular non-overlapping patches. Then, the encoder embeds the patches by linear projection with added positional embeddings. The combined embeddings are then processed via a series of self-attention-based Transformer blocks (Vaswani et al., 2017). We also consider a typical CNN architecture, *i.e.*, ResNet-50 (He et al., 2016), to demonstrate the versatility of MFM. To this end, we simply send the filtered spatial image to the CNN encoder as input.

**MFM decoder.** The decoder accomplishes the frequency reconstruction task. It can be of arbitrary form as long as its input is compatible with the encoder's output. Here, we simply adopt a lightweight linear layer as our decoder for efficiency, after which we perform FFT to convert each output image into the frequency domain for frequency reconstruction. The effect of different decoders is further studied in Appendix A.

**Reconstruction target.** Our MFM reconstructs the input by predicting the missing frequency values on the frequency spectrum. To faithfully recover the frequency values, we should define a frequency distance metric that considers both amplitude and phase as a loss function. Regarding each frequency value $\mathcal{F}(x)(u,v)$ as a two-dimensional Euclidean vector $\vec{f}$, one can easily derive that the magnitude of the vector corresponds to the amplitude while the angle corresponds to the phase. Inspired by Jiang et al. (2021), we thus define the frequency distance $\mathcal{D}(\cdot,\cdot)$ as the distance between the reconstructed vector $\vec{f}_r$ and the original vector $\vec{f}_o$ at each spectrum coordinate $(u,v)$:

$$
\begin{aligned}
\mathcal{D}\left(\vec{f}_r, \vec{f}_o\right) &= \left\| \vec{f}_r - \vec{f}_o \right\|_2^\gamma = \left| \mathcal{F}_r(x)(u,v) - \mathcal{F}_o(x)(u,v) \right|^\gamma \\
&= \left( (\mathcal{R}_r(x)(u,v) - \mathcal{R}_o(x)(u,v))^2 + (\mathcal{I}_r(x)(u,v) - \mathcal{I}_o(x)(u,v))^2 \right)^{\gamma/2},
\end{aligned}
\tag{4}
$$

where $\mathcal{R}(x)$ and $\mathcal{I}(x)$ are the real and imaginary part of $\mathcal{F}(x)$, respectively, $\gamma$ is an exponent to control the sharpness of the distance function and is set to 1 by default. For each image, the final loss function, *i.e.*, the average frequency distance of all spectrum positions can thus be written as:

$$
\mathcal{L} = \mathcal{D}\left(\mathcal{F}_r(x), \mathcal{F}_o(x)\right) = \frac{1}{HW} \sum_{u=0}^{H-1} \sum_{v=0}^{W-1} \left| \mathcal{F}_r(x)(u,v) - \mathcal{F}_o(x)(u,v) \right|^\gamma.
\tag{5}
$$

In practice, we compute the loss only on the masked area of the frequency spectrum instead of the full spectrum as the latter tends to decrease the accuracy according to our experiments.

## 3.2 RELATION WITH LOW-LEVEL IMAGE PROCESSING TASKS

The notion of recovering masked frequency components in MFM is reminiscent to the objectives in low-level image processing tasks, such as image super-resolution (SR), deblurring and denoising. In these tasks, a model takes a degraded image as input, and the aim is to restore the missing components. Different degradations corrupt different components in the frequency domain. As discussed in Section 1, for the image SR and deblurring tasks, most of the high-frequency components are removed while the low-frequency counterparts are retained; for the image denoising task, both low- and high-frequencies are significantly altered.

By analyzing the frequency spectrum of these tasks, we can observe how different frequencies of an image contribute to visual representation learning, thus gaining better insights on designing more effective learning objectives. Compared with these tasks, MFM provides a more general and unified frequency perspective to perform these low-level corruptions while being conceptually simpler: we directly remove certain frequencies on the frequency spectrum via a low-/high-pass filter. Our experiments show that MFM can achieve better performance than these tasks for representation learning. We will comprehensively study these tasks and show more details in the experiment section.

## 4 EXPERIMENTS

### 4.1 IMPLEMENTATION DETAILS

We use the vanilla ViT-Small (ViT-S/16), ViT-Base (ViT-B/16) and ResNet-50 models as the backbones in our study. We perform self-supervised pre-training on the ImageNet-1K (Deng et al., 2009) training set without labels. For ViT, our pre-training setting generally follows BEiT (Bao et al., 2022), while we *only* use random resized cropping ($224 \times 224$ resolution) and flipping as data

Table 1: **Ablations for MFM** with ViT-B/16 on ImageNet-1K. All models are based on 300-epoch pre-training, and we report top-1 fine-tuning accuracy. Unless specified, the default settings are: the mask type is random (*i.e.*, random sampling of low-/high-pass filters), the mask radius is 16, the mask shape is circle, the sampling ratio for low-pass filters is 50% (*i.e.*, 50% for low-pass filters and 50% for high-pass counterparts), the reconstruction target is masked frequencies on the spectrum, and the loss function is a frequency loss with $\gamma = 1$. Default entry is marked in ​ gray ​.

(a) **Mask type**. Random sampling of both filters works the best.

| Mask type | Top-1 acc (%) |
|-----------|---------------|
| none | 76.5 |
| low-pass | 82.4 |
| high-pass | 82.3 |
| random | **83.1** |

(b) **Mask radius**. Using a fixed radius is enough.

| Mask radius | Top-1 acc (%) |
|-------------|---------------|
| 8 | 82.8 |
| 16 | **83.1** |
| 24 | 82.7 |
| 32 | 82.6 |
| $[8, 24]$ | 83.0 |

(c) **Mask shape**. A circle mask is more accurate.

| Mask shape | Top-1 acc (%) |
|------------|---------------|
| circle | **83.1** |
| square | 82.9 |
| rhombus | 82.8 |

(d) **Sampling ratio**. Sampling low-/high-pass filters with an equal probability is effective.

| Sampling ratio | Top-1 acc (%) |
|----------------|---------------|
| 0.3 | 82.5 |
| 0.5 | **83.1** |
| 0.7 | 82.7 |

(e) **Reconstruction target**. Predicting only the masked frequencies yields better performance.

| Reconstruction target | Top-1 acc (%) |
|-----------------------|---------------|
| masked spectrum | **83.1** |
| full spectrum | 82.4 |

(f) **Loss function**. A frequency loss works better than spatial loss.

| Loss | Top-1 acc (%) |
|------|---------------|
| freq. ($\gamma = 1$) | **83.1** |
| freq. ($\gamma = 2$) | 82.5 |
| $\ell_1$ | 82.3 |
| $\ell_2$ | 82.2 |

augmentation, with dropout and stochastic depth not applied. We also do not use relative position or layer scaling. After pre-training, we conduct supervised end-to-end fine-tuning on ImageNet-1K image classification and ADE20K (Zhou et al., 2017) semantic segmentation to evaluate the quality of learned representations, following BEiT (Bao et al., 2022). For ResNet-50, we adopt the *same* pre-training configuration as that in ViT without further parameter tuning. We provide the detailed pre-training and fine-tuning recipes in Appendix G.

## 4.2    MAIN PROPERTIES

We start by ablating our MFM using ViT-B/16 as the default backbone. All experiments are conducted with 300-epoch pre-training and 100-epoch fine-tuning on the ImageNet-1K dataset unless otherwise specified. Several intriguing properties are observed.

**Masking strategy.** We first study different masking strategies on the frequency spectrum. We consider two kinds of filters: low-pass filter (*i.e.*, mask high frequencies), and high-pass filter (*i.e.*, mask low frequencies). As shown in Table 1a, masking and predicting either high frequencies ("low-pass" entry) or low frequencies ("high-pass" entry) perform significantly better than simply encoding and reconstructing the original image ("none" entry). This indicates that both high-frequency and low-frequency components are useful in representation learning, where the former largely depicts the object structure information such as outline or silhouette and the latter usually captures low-level statistics such as colors and styles. A random variant, *i.e.*, randomly selecting one filter from both low-pass and high-pass filters ("random" entry), benefits from all lens of frequencies, thus further improving the performance.

**Mask radius.** Table 1b studies the effect of mask radius, which controls the difficulty of our task. A larger radius leaves more frequencies for a low-pass filter while removes more frequencies for a high-pass filter. MFM works the best with a moderate difficulty. Using a fixed radius (*e.g.*, 16) performs slightly better than a random one, *i.e.*, the radius is uniformly sampled within a range (*e.g.*, $[8, 24]$).

**Mask shape.** We study three centrosymmetric mask shapes in Table 1c. Different mask shapes focus on different masking directions. Take low-pass filter as an example, a square shape removes more frequencies in the horizontal and vertical direction, while a rhombus one removes more in the diagonal direction. The results demonstrate that a circle mask shape that pays an equal attention to each direction on the frequency spectrum performs the best. We hypothesize that the effect of different mask shapes is largely correlated with the category statistics of pre-training datasets.

Table 2: **Comparison of SR, deblurring, denoising and MFM tasks** with ViT-B/16 on ImageNet-1K. All models are pre-trained for 300 epochs, and evaluated with top-1 fine-tuning accuracy. Corrupted image samples from ImageNet-1K training set with different degradation levels are visualized in both image and frequency domain. The studied hyper-parameter that controls the difficulty of degradation for each task is (a) downsampling scale factor, (b) Gaussian blur sigma, (c) Gaussian noise sigma, and (d) mask radius, respectively. More examples are provided in Appendix H. Zoom in for best view.

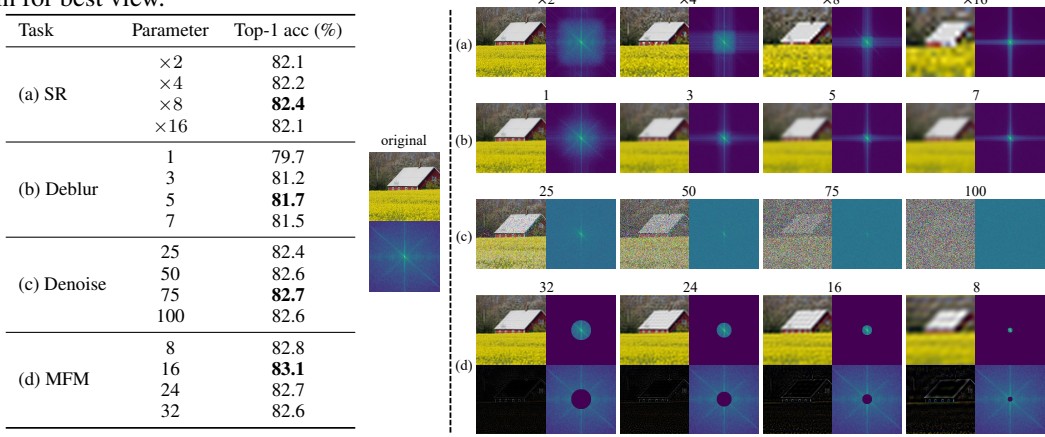

| Task | Parameter | Top-1 acc (%) |
|------|-----------|---------------|
| (a) SR | $\times 2$ | 82.1 |
|        | $\times 4$ | 82.2 |
|        | $\times 8$ | **82.4** |
|        | $\times 16$ | 82.1 |
| (b) Deblur | 1 | 79.7 |
|            | 3 | 81.2 |
|            | 5 | **81.7** |
|            | 7 | 81.5 |
| (c) Denoise | 25 | 82.4 |
|             | 50 | 82.6 |
|             | 75 | **82.7** |
|             | 100 | 82.6 |
| (d) MFM | 8 | 82.8 |
|         | 16 | **83.1** |
|         | 24 | 82.7 |
|         | 32 | 82.6 |

**Sampling ratio.** Table 1d ablates different sampling ratios for low-/high-pass filters. Here, the sampling ratio denotes the probability of sampling a low-pass filter, following a Bernoulli distribution. The results show that simply sampling both filters with an equal probability works the best.

**Reconstruction target.** Table 1e compares two reconstruction targets: 1) predicting *only* the masked frequencies on the frequency spectrum as in our default setting, and 2) recovering *both* the masked and unmasked frequencies on the frequency spectrum. Predicting the masked spectrum performs better than reconstructing the full spectrum by a clear margin (83.1% vs. 82.4%). This suggests that predicting the *invisible* signals is a more favourable task in representation learning, which is in accordance with the observation in recent MIM approaches.

**Loss function.** Table 1f studies the design of loss functions. A frequency loss (freq.) performs better than a spatial loss ($\ell_1$, $\ell_2$), with $\gamma = 1$ working the best. It makes sense as directly predicting the missing frequencies in the frequency domain better aligns to our MFM task.

### 4.3 DIAGNOSIS OF LOW-LEVEL IMAGE PROCESSING TASKS

In this subsection, we study the representation learning capability of low-level image processing tasks from a unified frequency perspective. We examine three representative tasks: image super-resolution (SR), deblurring, and denoising.

**Setup.** To ensure a direct comparison, we adopt the same pre-training and fine-tuning hyper-parameters as MFM and only alter the types of image degradation during pre-training. Specifically, for the SR task, we first use its standard data pre-processing, *i.e.*, bicubic downsampling, to downsample the input images by a scale factor. We then upsample them back to the original input size, *i.e.*, $224 \times 224$. For the deblurring task, we consider the commonly-used isotropic Gaussian filter and uniformly select the blur kernel size from $\{7, 9, 11, 13, 15, 17, 19, 21\}$ as suggested in Wang et al. (2021). For the denoising task, we employ the typical Gaussian noise. The intensity of both deblurring and denoising tasks is controlled by the standard deviation (*i.e.*, sigma value) of the Gaussian distribution. For all tasks, the reconstruction target is the original image but in the frequency domain via the same frequency loss as MFM.

**Observations.** Table 2 shows the results with different levels of degradation for each task. We first notice that the optimal degradation level of each task in the context of representation learning is much heavier than its original task setting. For instance, a standard SR task usually has a downsampling factor within $\times 4$, while we show that a much heavier $\times 8$ setting works the best. With right configuration of the task difficulty, all these tasks can achieve comparable or even better performance than their supervised counterpart (*e.g.*, 81.8% in Touvron et al. (2021a)), indicating that these low-level tasks are more or less helpful in representation learning. In addition, we observe that

representation learning benefits from all lens of frequencies. This can be verified by the superior performance of denoising over SR and deblurring. As visualized in the frequency spectrum of the example image, denoising tends to intensify all frequencies of the spectrum, while SR and deblurring only removes high-frequency components. Thus, the performance of denoising is much closer to MFM, as both utilize the full frequency spectrum. Attenuating and intensifying frequencies on the spectrum are essentially two different ways of performing corruption in the frequency domain. We believe other corruption types may also work well and leave this exploration for future work.

## 4.4 COMPARISON WITH PREVIOUS METHODS

### 4.4.1 IMAGE CLASSIFICATION

Table 3: ImageNet-1K top-1 fine-tuning accuracy of self-supervised models using ViT-S/16 and ViT-B/16 as the encoder. DINO and MoCo v3 use extra momentum encoder. BEiT requires extra 250M DALL-E data (Ramesh et al., 2021) to pre-train dVAE. BEiT and MAE also use mask tokens (inserted either in the encoder or the decoder). All entries are on an image size of $224 \times 224$. We use the actual processed images/views to measure the effective pre-training epochs (Zhou et al., 2022). Scratch indicates the supervised baseline in Touvron et al. (2021a). $^{\dagger}$: doubled attention heads. $^{\ddagger}$: our reproduced results with official code.

| Method | Pre-train data | Extra model | Mask token | Epochs | ViT-S | ViT-B |
|---|---|---|---|---|---|---|
| Scratch (Touvron et al., 2021a) | - | - | - | - | 79.9 | 81.8 |
| MoCo v3 (Chen et al., 2021) | IN-1K | momentum ViT | - | 600 | 81.4$^{\dagger}$ | 83.2 |
| DINO (Caron et al., 2021) | IN-1K | momentum ViT | - | 1600 | 81.5 | 82.8 |
| BEiT (Bao et al., 2022) | IN-1K+DALL-E | dVAE | ✓ | 300 | 81.3 | 82.9 |
| MAE (He et al., 2022) | IN-1K | - | ✓ | 300 | 80.6$^{\ddagger}$ | 82.9$^{\ddagger}$ |
| SR | IN-1K | - | - | 300 | 80.8 | 82.4 |
| Deblur | IN-1K | - | - | 300 | 79.4 | 81.7 |
| Denoise | IN-1K | - | - | 300 | 81.1 | 82.7 |
| MFM | IN-1K | - | - | 300 | 81.6 | 83.1 |

Table 4: ImageNet-1K top-1 fine-tuning accuracy of self-supervised models using ResNet-50 as the encoder. Table is split to three sub-tables for better placement. Results for other methods are taken from Fang et al. (2022) as we adopt the same fine-tuning recipe. $^{\dagger}$: modified ResNet-50 architecture.

(a) Training-from-scratch baselines.

| Method | Epochs | Top-1 acc (%) |
|---|---|---|
| Original$_{90}$ | - | 75.3 |
| PyTorch$_{90}$ | - | 76.1 |
| FixRes$_{120}$ | - | 77.0 |
| DeiT$_{300}$ | - | 78.4 |
| ResNet-RS$_{350}^{\dagger}$ | - | 78.8 |
| FAMS$_{400}$ | - | 79.5 |

(b) Fine-tuning for 100 epochs.

| Method | Epochs | Top-1 acc (%) |
|---|---|---|
| RSB A3 | - | 78.1 |
| SR | 300 | 77.9 |
| Deblur | 300 | 78.0 |
| Denoise | 300 | 77.5 |
| MFM | 300 | 78.5 |

(c) Fine-tuning for 300 epochs.

| Method | Epochs | Top-1 acc (%) |
|---|---|---|
| RSB A2 | - | 79.8 |
| SimSiam | 400 | 79.1 |
| MoCo v2 | 400 | 79.6 |
| SimCLR | 800 | 79.9 |
| BYOL | 400 | 80.0 |
| SwAV | 600 | 80.1 |
| MFM | 300 | 80.1 |

**ViT.** In Table 3, we compare the ImageNet-1K end-to-end fine-tuning results of self-supervised ViT-S/16 and ViT-B/16 models. We fine-tune ViT-S/16 for 200 epochs, and ViT-B/16 for 100 epochs. Other self-supervised models use the same or longer fine-tuning schedule. Compared with other representative self-supervised learners, our MFM can achieve comparable performance with fewer pre-training epochs while using *none* of the following: (i) extra data, (ii) extra model, (iii) mask token. This demonstrates the great potential of masked frequency modeling.

**ResNet-50.** We demonstrate that MFM can also pre-train a high-capacity ResNet-50 model. We simply adopt the same pre-training settings as ViT. During fine-tuning, we generally follow the advanced vanilla ResNet "training from scratch" recipe in RSB (Wightman et al., 2021) except that we use the AdamW optimizer (Loshchilov & Hutter, 2017) following Fang et al. (2022). Table 4 shows the results. Different from ViT, we observe performance degeneration of low-level image processing tasks like SR, deblurring and denoising compared with the RSB training-from-scratch baseline (Table 4b). We hypothesize this discrepancy is due to the architectural difference between ViT and CNN. Compared with ViT, the convolution operation in CNN tends to be more effective in capturing high-frequency components. Thus, encouraging a CNN model to reconstruct high-frequency

components of images brings no benefits to the performance. Instead, learning high-frequency information can compensate for the ability of ViT models in capturing the high-frequency components. In contrast, our MFM outperforms its supervised counterparts in both ViT and CNN architectures as it leverages both low- and high-frequency components. Even under a demanding training procedure, *e.g.*, fine-tuning for 300 epochs (Table 4c), MFM can still improve the supervised RSB A2 baseline by 0.3% and surpass several representative contrastive-based self-supervised learning methods.

### 4.4.2  SEMANTIC SEGMENTATION

We evaluate MFM and low-level image processing tasks on the ADE20K semantic segmentation benchmark. We use UperNet (Xiao et al., 2018) and adopt the same setup following BEiT (Bao et al., 2022). All models are fine-tuned for 160K iterations with an input resolution of $512 \times 512$. As shown in Table 5, our corruption-based models can achieve competitive performance compared with other representative self-supervised learners that are usually more expensive to compute.

Table 5: ADE20K semantic segmentation (mIoU) of ViT-B/16 models.

| Method | mIoU |
|---|---|
| Supervised (Touvron et al., 2021a) | 45.3 |
| MoCo v3 (Chen et al., 2021) | 47.2 |
| DINO (Caron et al., 2021) | 46.8 |
| BEiT (Bao et al., 2022) | 47.7 |
| MAE (He et al., 2022) | 48.1 |
| SR | 48.5 |
| Deblur | 47.0 |
| Denoise | 47.6 |
| MFM | 48.6 |

Table 6: **Robustness evaluation on six robustness benchmarks.** We report top-1 accuracy of ViT-B/16 (*left*) and ResNet-50 (*right*) models except for IN-C that uses the mean corruption error (mCE). The original ImageNet top-1 fine-tuning results are also appended for reference. The best results are in **bold**, and the second best results are underlined.

| Method | Robustness benchmarks | | | | | | Orig. | Method | Robustness benchmarks | | | | | | Orig. |
|---|---|---|---|---|---|---|---|---|---|---|---|---|---|---|---|
| | FGSM | PGD | IN-C (↓) | IN-A | IN-R | IN-SK | | | FGSM | PGD | IN-C (↓) | IN-A | IN-R | IN-SK | |
| Scratch | 46.3 | 21.2 | 48.5 | 28.1 | 44.7 | 32.0 | 81.8 | Scratch | **20.2** | **3.4** | 77.0 | 6.6 | 36.0 | 25.0 | 78.1 |
| MAE | 38.9 | 11.2 | 52.3 | 31.5 | 48.3 | 33.8 | 82.9 | SimMIM | 16.8 | 2.1 | 77.0 | 5.7 | 34.9 | 24.2 | 77.7 |
| SR | 46.1 | 21.5 | **46.3** | 29.1 | **49.2** | **35.5** | 82.4 | SR | 17.2 | 1.9 | **73.6** | 6.5 | 35.8 | 25.4 | 77.9 |
| Deblur | 42.5 | 17.2 | 49.2 | 25.3 | 46.9 | 33.2 | 81.7 | Deblur | 17.2 | 2.0 | 74.8 | 8.2 | **37.2** | 26.5 | 78.0 |
| Denoise | 47.6 | 24.3 | 47.8 | 30.7 | 48.4 | 34.8 | 82.7 | Denoise | 15.8 | 1.8 | 78.0 | 7.2 | 35.6 | 24.7 | 77.5 |
| MFM | **47.7** | **24.4** | 47.5 | **32.7** | 48.6 | 34.8 | **83.1** | MFM | 18.5 | 2.3 | 74.2 | **9.0** | 36.9 | **26.7** | **78.5** |

### 4.5  ROBUSTNESS EVALUATION

We evaluate the robustness of our models on a series of benchmarks in three aspects: (i) adversarial robustness, (ii) common corruption robustness, and (iii) out-of-distribution robustness. For (i), we study the adversarial examples generated by white-box attackers (*e.g.*, FGSM (Goodfellow et al., 2014) and PGD (Madry et al., 2017)) on ImageNet-1K validation set as well as natural adversarial examples on ImageNet-A (Hendrycks et al., 2021b); for (ii), we evaluate on ImageNet-C (Hendrycks & Dietterich, 2019) that includes 15 types of algorithmically generated corruptions with five levels of severity; for (iii), we test on ImageNet-R (Hendrycks et al., 2021a) and ImageNet-Sketch (Wang et al., 2019) that contain images with naturally occurring distribution shifts. We evaluate the same models fine-tuned on original ImageNet-1K (ViT-B/16 in Table 3 and ResNet-50 in Table 4b) without any specialized fine-tuning on the different validation sets. As shown in Table 6, we can conclude three observations: 1) Transformer-based models (*e.g.*, ViT) are more robust than the CNN counterparts (*e.g.*, ResNet-50). 2) Corruption-based tasks (*e.g.*, SR, Deblur, Denoise and MFM) are generally more robust than the MIM task (*e.g.*, MAE and SimMIM). 3) MFM achieves the best trade-off between standard performance and robustness (the robustness of MFM always ranks within the top two, while the standard accuracy is the best).

## 5  CONCLUSION

In this work, we have studied the effectiveness of low-level image processing tasks for visual representation learning from a new frequency perspective and introduced a unified, flexible and robust self-supervised visual pre-training framework to perform image corruptions in the frequency domain. We show that without relying on mask tokens or more complex designs (*e.g.*, discrete visual tokens), a simple *mask-frequency* strategy can achieve competitive performance for both ViT and CNN. We hope our unique frequency perspective can motivate the community to rethink the role of low-level tasks for unsupervised representation learning.

ETHICS STATEMENT

The proposed method learns statistics of the training dataset and may reflect the biases in the data. Debiased measures thus have to be taken. The method may be deployed with large-scale models and data, causing negative impacts on the environment.

REPRODUCIBILITY STATEMENT

We provide detailed hyper-parameter specifications for our experiments in the main text (Section 4) and the supplementary material (Appendix G) to ensure reproducibility. Code and models will be released at `https://www.mmlab-ntu.com/project/mfm/index.html` to facilitate future research.

ACKNOWLEDGEMENTS

This work is supported by NTU NAP, MOE AcRF Tier 2 (MOE-T2EP20120-0001, MOE-T2EP20221-0012), and under the RIE2020 Industry Alignment Fund – Industry Collaboration Projects (IAF-ICP) Funding Initiative, as well as cash and in-kind contribution from the industry partner(s).

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

# A    MORE ABLATIONS

Table 7: **More Ablations for MFM on ImageNet-1K.** All models are pre-trained for 300 epochs, and evaluated with top-1 fine-tuning accuracy. Default entry (the same settings as in Table 1 of the main text) is marked in  gray .

(a) **Decoder depth.**    A simple linear layer performs the best with lower training costs (Transformer blocks have a hidden size of 384 with 12 heads).

| Decoder | Blocks | Top-1 acc (%) |
|---------|--------|---------------|
| linear | - | **83.1** |
| Transformer blocks | 1 | 83.0 |
|  | 2 | 83.1 |
|  | 4 | 83.1 |
|  | 8 | 83.1 |

(b) **Masking in different domains.** Frequency masking is a more flexible and unified option for different architectures.

| Arch. | Task | Top-1 acc (%) |
|-------|------|---------------|
| ViT-B/16 | MIM | 82.8 |
|  | MFM | **83.1** |
| ResNet-50 | MIM | 77.7 |
|  | MFM | **78.5** |

**Decoder depth.** Table 7a ablates the effect of different decoders. MFM can work the best with a simple linear layer while benefiting from lower training costs compared with a deeper decoder.

**Frequency masking vs. spatial masking.** Considering that different design choices in existing masked image modeling (MIM) methods could affect the performance significantly, to eliminate the interference of other factors, we directly replace our frequency-domain-based masking with spatial-domain-based random patch masking for a fair comparison. The results are shown in Table 7b. Applying MIM to ResNet-50 leads to inferior performance even than the supervised baseline (*i.e.*, 78.1% in RSB A3 (Wightman et al., 2021)). In contrast, MFM outperforms the supervised RSB baseline as well as the MIM counterpart regardless of architectures, demonstrating that frequency masking is indeed a more flexible and unified option for different architectures.

# B    TRAINING TIME COMPARISON

We measure the pre-training time per epoch in Table 8. Note that MoCo v3 (Chen et al., 2021) and DINO (Caron et al., 2021) need to switch two global views and have four and 14 forward passes in total, respectively. BEiT (Bao et al., 2022), MAE (He et al., 2022) and MFM are 1-view methods without switching. MFM is relatively efficient compared with other MIM methods (*e.g.*, BEiT) except for MAE. However, taking only visible patches as input breaks the regular 2D structure of images, which makes MAE only applicable to ViT. In contrast, MFM is agnostic to architectures and can be flexibly applied for both ViT and CNN families. Considering the flexibility and universality of MFM, a slightly increasing time over MAE is acceptable.

Table 8: **Training time comparison.** The time is measured on the same 8-GPU machine with the same batch size using ViT-B/16, counted in relative to our approach. [†]: BEiT requires an additional stage to pre-train dVAE, which is not included.

| Method | Setup | Time per epoch |
|--------|-------|----------------|
| MoCo v3 | 2-view, 4-pass | 1.84× |
| DINO | (2+10)-view, 14-pass | 2.04× |
| BEiT | 1-view, 2-pass | 1.53×[†] |
| MAE | 1-view, 1-pass | 0.82× |
| MFM | 1-view, 1-pass | 1.00× |

# C    COMBINATION OF LOW-LEVEL IMAGE PROCESSING TASKS

In Table 1a of the main text, we have shown that randomly sampling low-/high-pass filters to predict both high and low frequencies benefits as MFM can make full use of the frequency spectrum, thus

leading to better performance. Here, we further study the effect of combining low-level image processing tasks, *i.e.*, SR, deblurring and denoising. Table 9 reports the ImageNet-1K top-1 fine-tuning accuracy of ViT-B/16 models. We find that combining these low-level corruptions does not bring similar benefits as MFM and even degrades the performance.

Table 9: **Combination of SR, deblurring, denoising tasks** using ViT-B/16 on ImageNet-1K. All models are pre-trained for 300 epochs, and evaluated with top-1 fine-tuning accuracy.

| Task | Top-1 acc (%) |
|---|---|
| *Individual task:* | |
| SR | 82.4 |
| Deblur | 81.7 |
| Denoise | **82.7** |
| *Integrated task:* | |
| SR+Denoise | 82.2 |
| Deblur+Denoise | 82.5 |

## D  FURTHER DISCUSSION

**Mask tokens.** In MIM methods, mask tokens are learnable patch embeddings inserted in the position where the input tokens are masked out. They are highly coupled with the Transformer architecture and not directly applicable to CNNs. Introducing special mask tokens in any intermediate stage of CNN is infeasible, as the intrinsic dense-sliding-window paradigm in convolution layers brings information leakage between visual features in previous layers (Fang et al., 2022). Thus, the large CNN family cannot directly benefit from this pre-training scheme like Transformers. In contrast, our MFM performs masking in the frequency domain without relying on mask tokens. Thus, MFM is agnostic to the architectures and can be flexibly applied for broader ViT and CNN families.

Table 10: **System-level comparison with Siamese-based hybrid MIM methods** (*e.g.*, iBOT (Zhou et al., 2022) and data2vec (Baevski et al., 2022)) using ViT-B/16 on ImageNet-1K. For a fair comparison, we re-implement iBOT without multi-crop augmentation (but keeping the two global views) and data2vec (Baevski et al., 2022) without additional losses of intermediate Transformer layers using their official code. Training costs are counted in relative to our approach. Note that MFM is agnostic to architectures while these hybrid methods are not.

| Method | Pre-text task | #Views | #Epochs | Top-1 acc (%) | Training costs |
|---|---|---|---|---|---|
| iBOT | MIM+CL | 2 | 300 | 82.0 | 2.14× |
| data2vec | MIM+CL | 2 | 300 | 83.0 | 1.60× |
| MFM | MFM | 1 | 300 | 83.1 | 1.00× |

**Comparison with hybrid MIM methods.** A line of recent research (Zhou et al., 2022; Baevski et al., 2022; Assran et al., 2022) combines MIM with contrastive learning (CL) into a Siamese framework and achieves better performance than a single task. Apart from adopting a Siamese network, additional techniques used in these works include multi-crop augmentation in Zhou et al. (2022); Assran et al. (2022) and multiple losses of intermediate Transformer layers in Baevski et al. (2022), without which their performance will be degraded significantly as shown in Table 10 (we take iBOT and data2vec as examples here since most of these works are concurrent to ours). Therefore, to eliminate the interference of other design factors, we mainly compare with pure MIM methods in our study. More advanced techniques used in these works may also be incorporated into MFM to further improve the performance, which is beyond the focus of this work.

**Concurrent work.** A concurrent work (Liu et al., 2022) also involves self-supervised learning in the frequency domain. Our work differs from theirs in the following aspects: 1) It aims at improving upon existing MIM approaches by additionally designing a more complex frequency decoder and computing losses in both spatial and frequency domain. In contrast, we aim at exploring an alternative frequency corruption strategy beyond MIM. Our method does not rely on any existing MIM

approaches and we show that MFM can also work well independently. 2) It is based on MAE (He et al., 2022) and still performs spatial masking with mask tokens. Thus, it is still not applicable to CNNs, whereas our frequency-domain-based masking strategy is agnostic to architectures. 3) As opposed to Liu et al. (2022) that mainly targets at improving MIM performance, we comprehensively study the effectiveness of low-level image processing tasks for representation learning from a unified frequency perspective and provide rather different insights that other low-level tasks beyond MIM can also work well.

**Limitations.** Our study has several limitations: 1) We mainly focus on the architectural flexibility and universality of MFM, while leaving the scaling behaviour under-explored. 2) We mainly evaluate the quality of learned representations on representative benchmarks, *i.e.*, ImageNet-1K image classification and ADE20K semantic segmentation, following Bao et al. (2022). The transferability on more downstream tasks can be studied in future research. 3) Despite the intriguing properties of the Fourier domain, the redundancy in frequencies may still exist. More advanced information suppression strategies can be further explored. We believe that our MFM can also complement contrastive learning and MIM approaches to further improve the performance. We leave these explorations for future work.

**Future work.** In this paper, we have shown that MFM is a simple, unified and flexible self-supervised pretext task for various architectures. Compared with MIM, MFM can embrace broader architectures (*e.g.*, ViT, CNN, *etc.*) and has more appealing robustness. Some possible future research directions may include: 1) More self-supervised learning works in the frequency domain with different modalities (*e.g.*, image, video, audio, *etc.*). 2) Combine MFM with existing contrastive learning and MIM paradigms to further improve the performance. 3) Apply MFM for model robustness analysis and calibration. 4) The idea of MFM may also be used in low-level image reconstruction and synthesis tasks.

# E PSEUDOCODE

---

**Algorithm 1** Pseudocode of MFM in a PyTorch-like style.

---

```
# f: backbone encoder (e.g., vit, cnn) + linear prediction head
# mask: frequency mask of low-/high-pass filters sampled with a Bernoulli
    distribution, i.e., mask = Bernoulli(p) ? m : 1 - m (m is defined in Eq. (2),
    p is the probability of sampling a low-pass filter m)
# gamma: exponent to control the sharpness of the frequency distance

for (x, mask) in loader: # load a minibatch x with N samples
    x = aug(x) # random view, NxCxHxW
    # convert spatial domain into frequency domain
    x_freq = fft2(x) # 2D FFT
    x_freq = fftshift(x_freq, dim=(-2, -1)) # shift low frequency to the center
    x_freq = x_freq * mask # mask a portion of frequencies
    x_freq = ifftshift(x_freq, dim=(-2, -1)) # restore the original frequency order
    # convert frequency domain back into spatial domain
    x_corrupted = ifft2(x_freq).real # 2D iFFT (only keep the real part)
    x_predicted = f(x_corrupted) # predicted view, NxCxHxW

    loss = FrequencyLoss(x_predicted, x, gamma) # frequency loss
    # only compute the frequency loss on the masked area
    loss = (loss * (1 - mask)).sum() / (1 - mask).sum()

    # update model
    loss.backward()
    update(f)

def FrequencyLoss(x, y, gamma):
    x_freq, y_freq = fft2(x), fft2(y) # 2D FFT
    # shift low frequency to the center
    x_freq = fftshift(x_freq, dim=(-2, -1))
    y_freq = fftshift(y_freq, dim=(-2, -1))
    # stack the real and imaginary parts along the last dimension
    x_freq = stack([x_freq.real, x_freq.imag], -1)
    y_freq = stack([y_freq.real, y_freq.imag], -1)
    # compute the frequency distance
    d = (x_freq - y_freq) ** 2
    return (d[..., 0] + d[..., 1]) ** (0.5 * gamma)
```

---

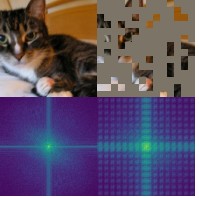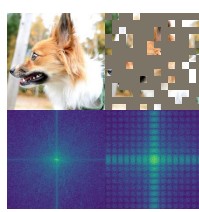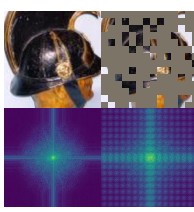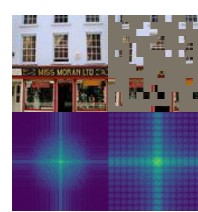

Figure 3: **Example frequency spectrums of spatial-domain-based random patch masking** from ImageNet-1K training set. The masking ratio is 75%. Performing patch-wise masking in the spatial domain incurs grid-wise artifacts on the frequency spectrum.

## F    FREQUENCY SPECTRUM OF MIM

Figure 3 visualizes some example frequency spectrums of spatial-domain-based random patch masking used in MIM. Performing patch-wise masking in the spatial domain incurs grid-wise artifacts on the frequency spectrum, preventing further meaningful observations.

## G    IMPLEMENTATION DETAILS

**Pre-training.** Table 11 summarizes the pre-training settings for vanilla ViT and ResNet-50 models. All experiments are conducted on 16 V100 32G GPUs for ViT models and 8 V100 32G GPUs for ResNet-50. The configurations are *shared* by different architectures, without specialized tuning. This demonstrates that MFM is *general* across architectures.

**Fine-tuning.** Table 12 and Table 13 summarize the fine-tuning settings for vanilla ViT and ResNet-50 models, respectively. The configurations for ViT are *shared* across models, except that smaller models are fine-tuned longer. The configurations for ResNet-50 basically follow Wightman et al. (2021), except that we adopt the AdamW optimizer following Fang et al. (2022).

**Semantic segmentation on ADE20K.** We use UperNet (Xiao et al., 2018) following the configurations in BEiT (Bao et al., 2022). Specifically, we use AdamW as the optimizer and fine-tune for 160K iterations with a batch size of 16. We search the learning rate for all the results in Table 5 of the main text. The input resolution is $512 \times 512$, and we use single-scale inference. As suggested in BEiT (Bao et al., 2022), we initialize all segmentation models using model weights after supervised fine-tuning on ImageNet-1K, following the common practice of BERT (Devlin et al., 2019) fine-tuning in NLP (Pruksachatkun et al., 2020).

Table 11: **Pre-training settings for vanilla ViT-S/16, ViT-B/16 and ResNet-50 models on ImageNet-1K.** Note that we adopt the *same* pre-training configurations across different architectures without further parameter tuning.

| Configuration | Value |
|---|---|
| Optimizer | AdamW (Loshchilov & Hutter, 2017) |
| Pre-training epochs | 300 |
| Peak learning rate | 1.2e-3 |
| Batch size | 2048 |
| Weight decay | 0.05 |
| Optimizer momentum | $\beta_1, \beta_2 = 0.9, 0.95$ (Chen et al., 2020a) |
| Learning rate schedule | Cosine decay |
| Warmup epochs | 20 |
| Gradient clipping | 3.0 |
| Dropout (Srivastava et al., 2014) | ✗ |
| Stochastic depth (Huang et al., 2016) | ✗ |
| LayerScale (Touvron et al., 2021b) | ✗ |
| Data augmentation | RandomResizedCrop |
| Pos. emb. in Transformer layers | 1-D absolute pos. emb. (Dosovitskiy et al., 2020) |
| Patch size | 16 |
| Pre-training resolution | 224 |

Table 12: **Fine-tuning settings for vanilla ViT-S/16 and ViT-B/16 on ImageNet-1K.** We fine-tune ViT-S/16 for 200 epochs, and ViT-B/16 for 100 epochs. All other hyper-parameters are the same.

| Configuration | Value |
|---|---|
| Optimizer | AdamW (Loshchilov & Hutter, 2017) |
| Fine-tuning epochs | 200 (S), 100 (B) |
| Peak learning rate | 8e-3 |
| Layer-wise learning rate decay (Bao et al., 2022) | 0.8 (Clark et al., 2020) |
| Batch size | 2048 |
| Weight decay | 0.05 |
| Optimizer momentum | $\beta_1, \beta_2 = 0.9, 0.999$ |
| Learning rate schedule | Cosine decay |
| Warmup epochs | 5 |
| Loss function | Cross-entropy loss |
| Gradient clipping | ✗ |
| Dropout (Srivastava et al., 2014) | ✗ |
| Stochastic depth (Huang et al., 2016) | 0.1 |
| Mixup (Zhang et al., 2017a) | 0.8 |
| Cutmix (Yun et al., 2019) | 1.0 |
| Label smoothing (Szegedy et al., 2016) | 0.1 |
| Random augmentation (Cubuk et al., 2020) | 9 / 0.5 |
| Patch size | 16 |
| Fine-tuning resolution | 224 |
| Test resolution | 224 |

Table 13: **Fine-tuning settings for vanilla ResNet-50 on ImageNet-1K.** The hyper-parameters generally follow Wightman et al. (2021), except that we adopt the AdamW optimizer following Fang et al. (2022).

| Configuration | 100 epoch FT | 300 epoch FT |
|---|---|---|
| Optimizer | AdamW (Loshchilov & Hutter, 2017) | |
| Peak learning rate | 12e-3 | |
| Layer-wise learning rate decay (Bao et al., 2022) | ✗ | |
| Batch size | 2048 | |
| Weight decay | 0.02 | |
| Learning rate schedule | Cosine decay | |
| Warmup epochs | 5 | |
| Loss function | Binary cross-entropy loss | |
| Gradient clipping | ✗ | |
| Dropout (Srivastava et al., 2014) | ✗ | |
| Stochastic depth (Huang et al., 2016) | ✗ | |
| Mixup (Zhang et al., 2017a) | 0.1 | |
| Cutmix (Yun et al., 2019) | 1.0 | |
| Label smoothing (Szegedy et al., 2016) | 0.1 | ✗ |
| Repeated augmentation (Berman et al., 2019; Hoffer et al., 2019) | ✗ | ✓ |
| Random augmentation (Cubuk et al., 2020) | 6 / 0.5 | 7 / 0.5 |
| Fine-tuning resolution | 160 | 224 |
| Test resolution | 224 | |
| Test crop ratio | 0.95 | |

# H  VISUALIZATION

We provide more qualitative results of corrupted images in Figure 4 following Table 2 in the main text, as well as recovered images using unseen ImageNet-1K (Figure 5) and COCO (Figure 6) *validation* images.

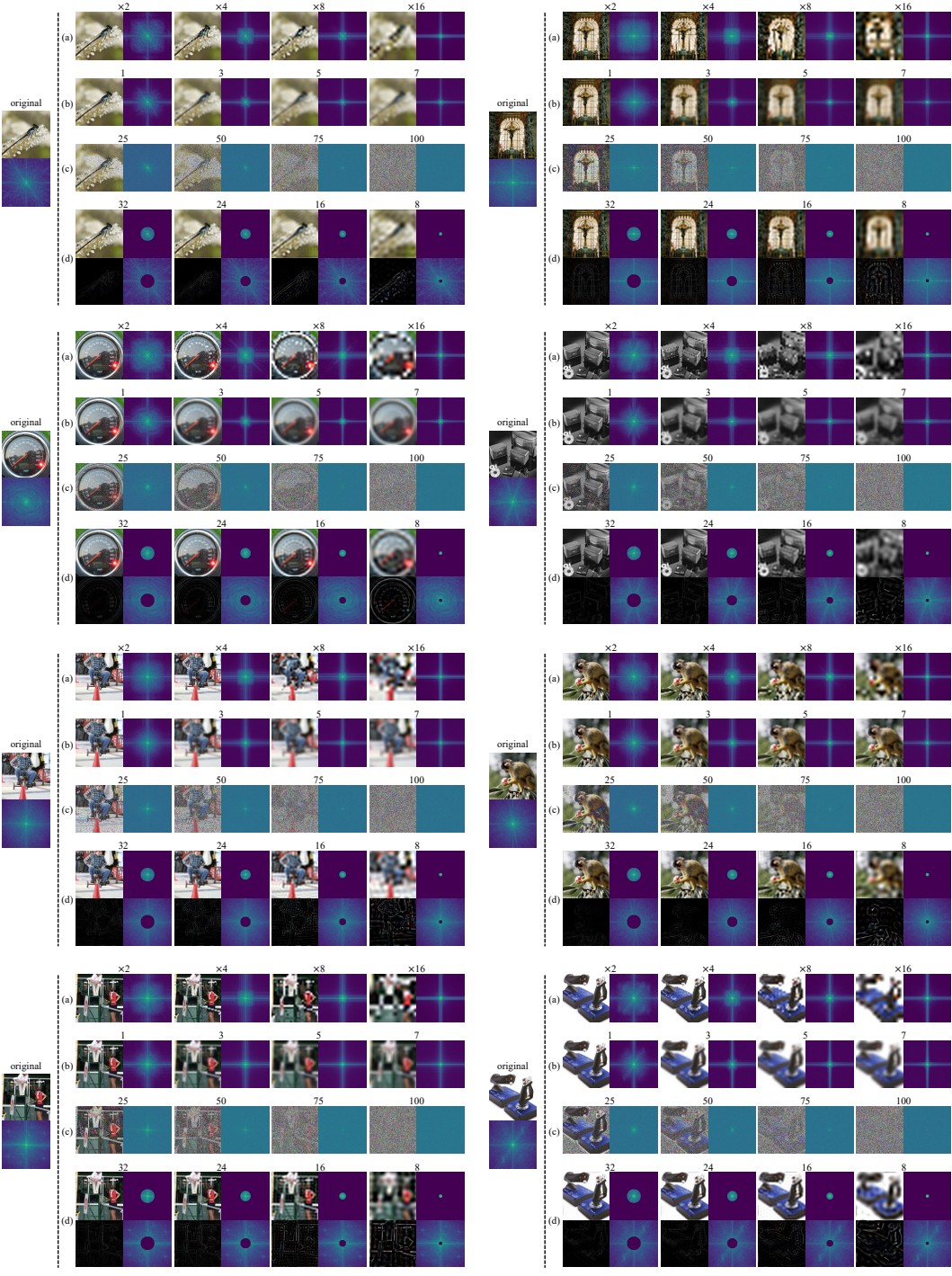

Figure 4: **More visualizations of corrupted image samples** from ImageNet-1K training set following Table 2 in the main text. We visualize both images and their frequency spectrums with different degradation levels. (a) SR, (b) Deblur (kernel size 21), (c) Denoise, (d) MFM. Each task achieves its best performance with a moderate degradation intensity. See Section 4.3 in the main text for more discussion. Zoom in for best view.

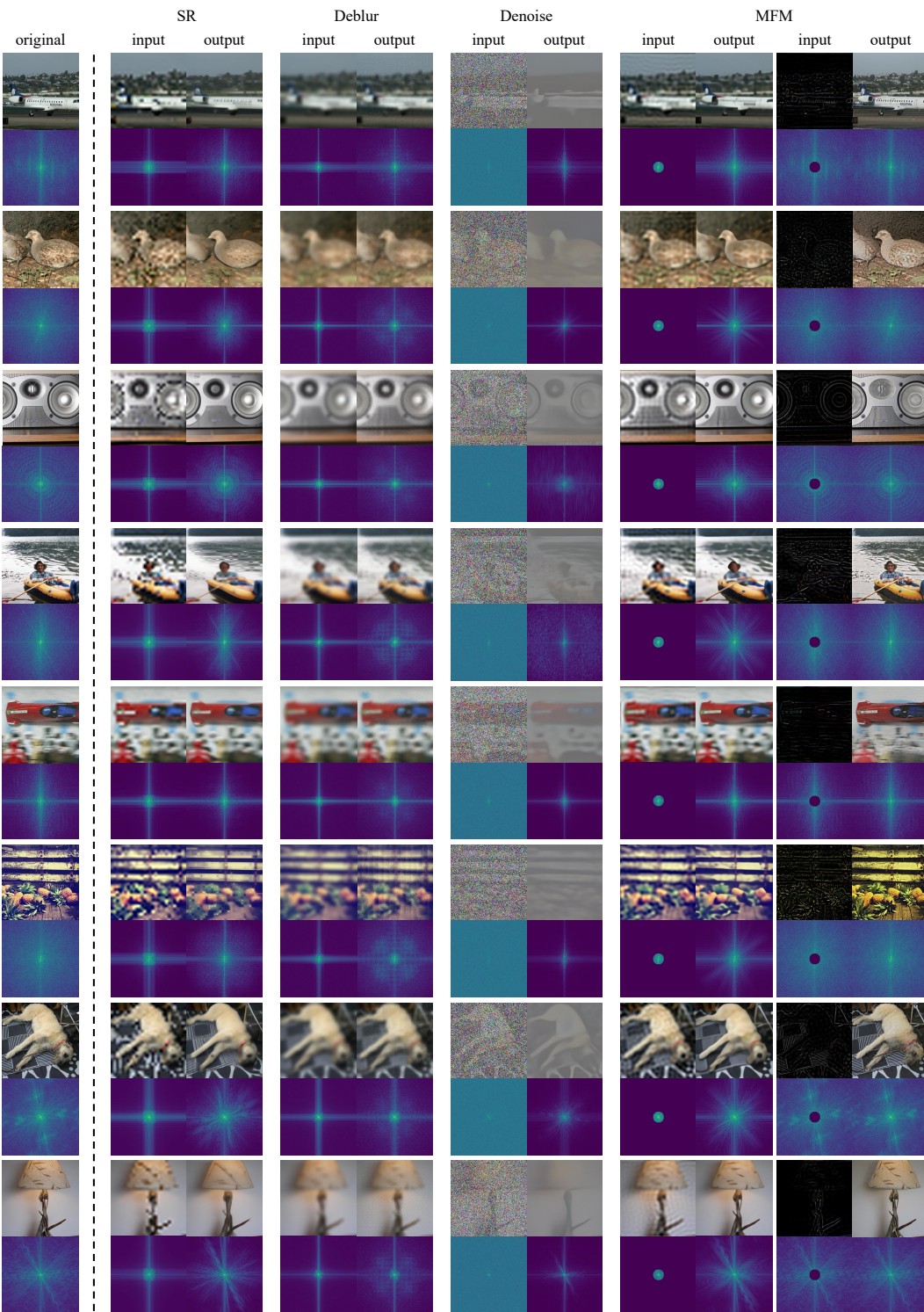

Figure 5: **Example results of recovered images** on ImageNet-1K *validation* set for SR, deblurring, denoising and MFM tasks. We visualize both images and their frequency spectrums. We use the best pre-trained model of each task in Table 2 of the main text for visualization, *i.e.*, the downsampling scale factor is ×8 for SR, the Gaussian blur sigma is 5 for Deblur, the Gaussian noise sigma is 75 for Denoise, and the mask radius is 16 for MFM[†]. Compared with SR, Deblur and Denoise, MFM can utilize both high-frequency and low-frequency information for prediction. Zoom in for best view.

[†]As MFM only predicts the masked area of the frequency spectrum, we overlay the output with the visible frequency spectrum for better visual quality.

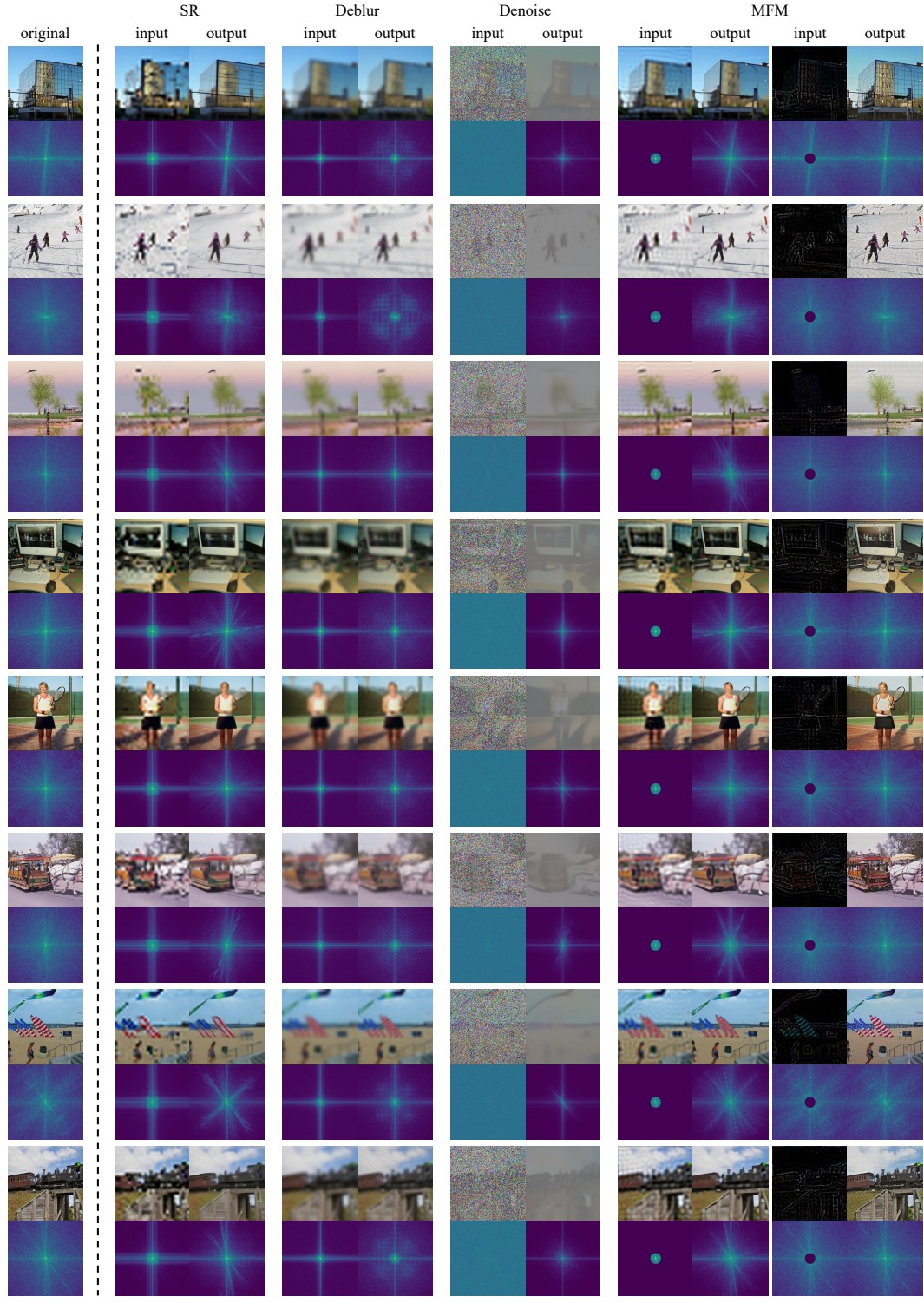

Figure 6: **Example results of recovered images** on COCO *validation* set for SR, deblurring, denoising and MFM tasks, using the models pre-trained on ImageNet-1K (the same model weights as in Figure 5). We visualize both images and their frequency spectrums. Zoom in for best view.

