# OpenReview forum: "Masked Frequency Modeling for Self-Supervised Visual Pre-Training"
_ICLR.cc/2023/Conference — ICLR 2023 poster_

### Official Review · Reviewer_Nxgg · 2022-10-17

**Confidence:** 3
**Correctness:** 3
**Technical Novelty And Significance:** 3
**Empirical Novelty And Significance:** 2
**Recommendation:** 6

**Clarity, Quality, Novelty And Reproducibility:**

## Clarity

As indicated in the weaknesses points above, I think the calrity of the paper could be improved. I was left with a number of questions regarding the figures and the overall flow of the MFM model. Addressing the concerns I raised above could solve most of these issues.

## Novelty

The idea appears to be sufficiently novel to my knowledge

## Quality

Aside from the clarity issues above and a few small grammar issues, the quality is adequate.

**Strength And Weaknesses:**

## Strengths

- The method is agnostic to architectures

- It is interesting that denoising is closer to MFM and the effect of noise also intensifies all frequencies of the spectrum. Would it be possible to compare with masking as a corruption as well to see what that does to the overall frequency of the image?

## Weaknesses

- Figure 1 is hard to interpret. I can see that there are differences between MFM and the applied corruptions, but the frequency domain has no axis labels and it is difficult to gain intuition behind why the frequency domain plots look different from the corruptions, and what significance this implies.

- Figure 2, which frequencies in the "mask frequencies" brackets in the figure are low and high frequencies? Or are they frequencies and amplitudes? What is the red circle in the first part of the figure? Is that the selected radius which is specified in equation 2?

- After equation 3, it says that the filtered images are "fed to and encoder as input, following a Bernoulli distribution." This statement is not clear, what does it mean?

- After equation 3, the authors state that the model still takes the spatial images as input as well. It would be good to state this earlier in the paper. Until this point, I was under the impression that the model would only receive the frequency domain as input.

- Given the confusion from the previous points, it would be nice to see a better diagram of the entire model architecture in a figure (maybe revamping figure 2) to allow the reader to visualize what the entire model flow looks like. It is not clear to me when and where the frequency inputs come into the model as inputs. For example, figure 2 looks like like the frequencies are input into the encoder, but the MFM Encoder section says nothing about this.

- Did you ever consider using difference unit circle norms as shapes for ablation c (cirlce = 2 norm, square = infinity norm, etc)? The reason I ask this is because the frequency domain charts seem to show some interesting designs which would be captured by unit cirlce norms < 1 which would form a star-like shape.

- Building on the point above, what is the shape of the rhombus in this context. Would this be equivalent to the 1-norm unit circle?

- page 8: "representation learning benefits from all lens of frequencies..." What does this mean?

- mask tokens are not used, which means that pretraining should be much more expensive than other MIM approahces, right? Because most MIM models produce a shorter sequence length to the transformer. Can you provide a breakdown of the pre-training per iteration computational cost in terms of wall-clock time between MFM and some baselines?

## Minor

Section 2 MIM section: "Besides, iGPT..." IT is not clear why besides is used here, besides what?

**Summary Of The Paper:**

The authors focus masking frequencies in the input image for self-supervised pretraining rather than masking out sections of the spatial domain. The insight is that the masked frequencies will carry more information about the patterns in the underlying image as compared to spatial patches.

**Summary Of The Review:**

Overall, I think the idea is interesting and novel. It is also intuitive that the frequency domain may capture a wider range of information that is covered by multiple corruptions. However, I would like to see some of the above mentioned clarity issues resolved.

Additionally, it is not clear to me why someone would choose to use this strategy. The bottom line results are somewhat ambiguous, and there are several recent SSL baselines which *could* be added that achieve higher performance.  Therefore I think it could be better to include more information analyzing the benefits of using MFM and possible future research directions which may be able to make further use of this work.

---

> ### Author Response · Authors · 2022-11-18
> **Response to Reviewer Nxgg (Part 2/2)**
>
> > **Q9:** Page 8: “representation learning benefits from all lens of frequencies...” What does this mean?
>
> **A9**: As observed in Table 2 (page 7) and analyzed in this paragraph (page 8), MFM and denoising achieve better performance than SR and deblurring since they corrupt both low and high frequencies on the frequency spectrum. This demonstrates that representation learning can benefit from predicting both low and high (i.e., all lens of) frequencies.
>
> > **Q10:** Mask tokens are not used, which means that pre-training should be much more expensive than other MIM approaches, right? Because most MIM models produce a shorter sequence length to the transformer. Can you provide a breakdown of the pre-training per iteration computational cost in terms of wall-clock time between MFM and some baselines?
>
> **A10:** We provide comparison of the pre-training time per epoch as suggested below (the time is measured on the same 8-GPU machine with the same batch size using ViT-B/16, counted in relative to our approach):
>
> | Method  | Setup                | Time per epoch             |
> |---------|----------------------|----------------------------|
> | MoCo v3 | 2-view, 4-pass       | 1.84x                      |
> | DINO    | (2+10)-view, 14-pass | 2.04x                      |
> | BEIT    | 1-view, 2-pass       | 1.53x (w/ extra tokenizer) |
> | MAE     | 1-view, 1-pass       | 0.82x                      |
> | MFM     | 1-view, 1-pass       | 1.00x                      |
>
> BEiT requires an additional stage to pre-train dVAE, whose training time is not included here. Please note that MoCo v3 and DINO need to switch two global views and have four and 14 forward passes in total, respectively. BEiT, MAE and MFM are 1-view methods without switching. MFM is an efficient method compared with other MIM approaches (e.g., BEiT) except for MAE. Although MAE is more efficient, taking only visible patches as input breaks the regular 2D structure of images, which makes MAE not applicable to other architectures like CNNs. In contrast, MFM is agnostic to architectures and can be flexibly applied to various architectures (e.g., ViT, CNN, etc.). Considering the universality and flexibility, a slightly increasing time over MAE is acceptable. We have added this comparison in Appendix B (page 14) of the revised manuscript.
>
> > **Q11:** Section 2 MIM section: “Besides, iGPT...”. It is not clear why besides is used here, besides what?
>
> **A11:** “Besides” here means, apart from the mask-patch strategy, there also exist other auto-regressive variants like iGPT.
>
> > **Q12:** It could be better to include more information analyzing the benefits of using MFM and possible future research directions which may be able to make further use of this work.
>
> **A12:** As analyzed and validated in the paper, MFM mainly has two advantages compared with MIM: (i) a flexible and unified approach applicable to broader architectures (e.g. ViT, CNN, etc.), and (ii) more appealing robustness. Some possible future research directions may include: (1) More self-supervised learning works in the frequency domain with different modalities (e.g., image, video, audio, etc.). (2) Combine MFM with existing contrastive learning and MIM paradigms to further improve the performance. (3) Apply MFM for model robustness analysis and calibration. (4) The idea of MFM may also be used in low-level image reconstruction and synthesis tasks. Thank you for the suggestion, and we have added more discussions in Appendix D (page 15-16) of the revised manuscript.

---

> > ### Comment · Reviewer_Nxgg · 2022-11-25
> > **Thank you for the responses**
> >
> > Thank you for the detailed responses. You have adequately addressed my concerns and I will raise my score accordingly.

---

> > > ### Author Response · Authors · 2022-11-25
> > > **Thank you for the feedback**
> > >
> > > We thank the reviewer for the positive feedback and raising the score. We are happy to hear that we were able to address your concerns.

---

> ### Author Response · Authors · 2022-11-18
> **Response to Reviewer Nxgg (Part 1/2)**
>
> Thank you for the constructive comments. Please find the following for our response.
>
> > **Q1:** Would it be possible to compare with masking as a corruption as well to see what that does to the overall frequency of the image?
>
> **A1:** Performing patch-wise masking in the spatial domain will incur grid-wise artifacts on the frequency spectrum. Thus, we cannot obtain any meaningful observations from its frequency spectrum. We have provided some corresponding visualizations in Appendix F (page 17) of the revised manuscript for reference.
>
> > **Q2:** Figure 1 is hard to interpret. I can see that there are differences between MFM and the applied corruptions, but the frequency domain has no axis labels and it is difficult to gain intuition behind why the frequency domain plots look different from the corruptions, and what significance this implies.
>
> **A2:** We have made Figure 1 clearer by adding the colorbar axis. Please note that all frequency spectrums are normalized between 0-1. For a frequency spectrum, the closer to the center, the lower the frequency. The main information we want to convey from Figure 1 is that the downsampling and blur operations dominantly remove the high-frequency components, while adding noises tends to intensify all frequencies. In contrast, MFM can remove frequencies in a more unified and flexible way as it directly performs masking in the frequency domain.
>
> > **Q3:** Figure 2, which frequencies in the “mask frequencies” brackets in the figure are low and high frequencies? Or are they frequencies and amplitudes? What is the red circle in the first part of the figure? Is that the selected radius which is specified in Equation 2?
>
> **A3:** In the “mask frequencies” brackets, the upper branch is the low-pass filtered frequency spectrum and its corresponding iFFT converted image, while the lower branch is the high-pass filtered frequency spectrum and its corresponding iFFT converted image. The frequency spectrums are visualized with the amplitudes. The red circle denotes the selected mask radius defined in Equation 2. Sorry for the confusion and we have made these points clearer in the caption of Figure 2 (page 4) in the revised manuscript.
>
> > **Q4:** After Equation 3, it says that the filtered images are “fed to an encoder as input, following a Bernoulli distribution.” This statement is not clear, what does it mean?
>
> **A4:** We randomly sample low-pass or high-pass filtered images as input to ensure that only one view is forwarded to the encoder at a time. The random sampling process follows the Bernoulli distribution. We have fixed this sentence for better understanding in the revised manuscript.
>
> > **Q5:** After Equation 3, the authors state that the model still takes the spatial images as input as well. It would be good to state this earlier in the paper. Until this point, I was under the impression that the model would only receive the frequency domain as input.
>
> **A5:** Thank you for the suggestion. We have highlighted this point by placing it at footnote 2 (page 4) in the revised manuscript.
>
> > **Q6:** Given the confusion from the previous points, it would be nice to see a better diagram of the entire model architecture in a figure (maybe revamping Figure 2) to allow the reader to visualize what the entire model flow looks like. It is not clear to me when and where the frequency inputs come into the model as inputs. For example, Figure 2 looks like the frequencies are input into the encoder, but the MFM Encoder section says nothing about this.
>
> **A6:** Thank you for the suggestion. We have revised the caption of Figure 2 (page 4), the “Masking strategy”, “MFM encoder” and “MFM decoder” sections (page 4-5) for clarity issues. We have also provided the pseudocode of MFM to reflect the entire workflow in Appendix E (page 16) of the revised manuscript.
>
> > **Q7:** Did you ever consider using different unit circle norms as shapes for ablation (c) (cirlce = 2 norm, square = infinity norm, etc.)? The reason I ask this is because the frequency domain charts seem to show some interesting designs which would be captured by unit circle norms < 1 which would form a star-like shape.
>
> **A7:** Thank you for providing this perspective. This is an interesting interpretation on different mask shapes. We have considered three commonly used shapes, i.e., circle (2-norm), square (infinity-norm), and rhombus (1-norm) in our paper. The reason why we do not consider a star-like shape is that there could be many possible star-like shapes when unit norm < 1. Here, we take unit norm = 0.5 as an example to study the star-like shape as suggested and its top-1 fine-tuning accuracy is 82.7%, which is lower than the three shapes (circle: 83.1%, square: 82.9, rhombus: 82.8%) we studied in the paper.
>
> > **Q8:** Building on the point above, what is the shape of the rhombus in this context. Would this be equivalent to the 1-norm unit circle?
>
> **A8:** Yes, it is equivalent to the 1-norm unit circle.

---

### Official Review · Reviewer_34gh · 2022-10-20

**Confidence:** 4
**Correctness:** 3
**Technical Novelty And Significance:** 2
**Empirical Novelty And Significance:** 3
**Recommendation:** 5

**Clarity, Quality, Novelty And Reproducibility:**

This paper has good overall clarity with several minor issues, marginal novelty due to straightforward method and interpretation, and good reproducibility with well-described training recipes.

**Strength And Weaknesses:**

Strengths:

1. The paper provides empirical evidence on the possibility of using various low-level image processing methods for self-supervised learning, including high-pass or low-pass filtering, denoising, super-resolution and deblurring, which will be a useful contribution to the community.

2. The proposed method could achieve comparable performance for ViT-S, ViT-B, Resnet50 and improved robustness for ViT-B.

3. The paper is generally well-written and easy to follow.

Weaknesses:

1. The major concern is that the methodology contribution is straightforward:
- The masked frequency modeling proposed is straightforward and limited to low-pass and high-pass filtering, which are basically blurring and sharpening. To be a principled method, it would be better to include more general forms of frequency manipulations, which would correspond to modeling more complex periodical patterns in the spatial domain.
- Although the authors attempt to provide a unified perspective of the method by relating to denoising, super-resolution and deblurring (Section 3.2), such perspective is very natural to have since most image processing methods in the spatial domain would have a corresponding interpretation in the frequency domain.

2. The authors claim that no extra model (e.g. momentum teacher) and no mask token are advantages of MFM over other self-supervised methods. However, these differences are more about implementation choices rather than principled advantages.

3. Although the experiments show promising results, they lack further analysis and insights:
- Why is no gain observed for CNNs?
- Why could masked frequency modeling achieve better robustness than MAE?
- etc.

Minor issues:
- In Figure 2, what does the FFT after the linear head mean? This seems to be contradicting the text.
- It is unclear how the linear head would output the frequency map of the same size as the original image, especially for CNNs encoder.
- In Eq.4, the lowercase symbols f_r and f_o seem to be redundant and unnecessary.
- In Table 1(f), are the l1 and l2 losses in spatial domain? This is not clearly stated.
- In Table 6, the robustness is not compared to other self-supervised methods for ResNet.


**Summary Of The Paper:**

This paper proposes a self-supervised representation learning method by applying high-pass or low-pass filters on the image and restoring the missing frequency components. The experiments also include results for self-supervised learning via other common image restoration methods such as denoising, super-resolution and deblurring.

**Summary Of The Review:**

This paper presents some interesting empirical results about using different image corruptions for self-supervised learning. However, the proposed method is currently not principled enough to be a significant technical contribution. I would recommend incorporating frequency analysis in a more general form beyond low-pass and high-pass filtering.

---

> ### Author Response · Authors · 2022-11-18
> **Response to Reviewer 34gh (Part 2/2)**
>
> > **Q6:** Minor issues: (1) In Figure 2, what does the FFT after the linear head mean? (2) It is unclear how the linear head would output the frequency map of the same size as the original image, especially for CNN encoder. (3) In Eq. 4, the lowercase symbols $\vec{f_r}$ and $\vec{f_o}$ seem to be redundant and unnecessary. (4) In Table 1(f), are the $l1$ and $l2$ losses in spatial domain? (5) In Table 6, the robustness is not compared to other self-supervised methods for ResNet.
>
> **A6:** (1) It means the output image from the linear head will be converted into the frequency domain with FFT so that we can compute the loss on the frequency spectrum. We have made this point clearer in the “MFM decoder” paragraph in Section 3.1 (page 5). (2) We use [`torch.nn.PixelShuffle`](https://pytorch.org/docs/stable/generated/torch.nn.PixelShuffle.html) operation in PyTorch to rearrange the output from the linear layer to ensure the same size as the original image. (3) We use lower case symbols $\vec{f_r}$ and $\vec{f_o}$ to represent the vectors in a more concise format. Using $F_r(x)(u, v)$ and $F_o(x)(u, v)$ will make the equation too long. (4) Yes, they are losses in the spatial domain. We have made it clearer in the “Loss function” paragraph in Section 4.2 (page 7). (5) For the ResNet-50 entry, MAE is infeasible since taking only visible patches as input breaks the regular 2D structure of images. Here, we use SimMIM that processes the full images as an alternative and results are listed below:
>
> | Method    | FGSM | PGD | IN-C (↓) | IN-A | IN-R | IN-SK | Orig. 	|
> |--------|:----:|:---:|:--------:|:----:|:----:|:-----:|:-----:|
> | SimMIM 	| 16.8 	| 2.1  |   77.0   |  5.7  | 34.9 |  24.2 	|  77.7 	|
> | MFM    	| **18.5** 	| **2.3**  |  **74.2**   |  **9.0**  | **36.9** |  **26.7** 	|  **78.5** 	|
>
> MFM is still more robust than its MIM counterpart. We have added these results in Table 6 (page 9) of the revised manuscript.

---

> ### Author Response · Authors · 2022-11-18
> **Response to Reviewer 34gh (Part 1/2)**
>
> Thank you for the constructive comments. Please find the following for our response.
>
> > **Q1:** It would be better to include more general forms of frequency manipulations.
>
> **A1:** We mainly choose low-pass and high-pass filters in our study since they are two representative operations in the frequency domain. In our preliminary experiments, we also tried other manipulations (e.g., masking a band frequency, attenuating the phase or amplitude, etc.). However, these more complex operations did not lead to better performance than simply adopting the low-/high-pass filtering. It is possible that there could exist more complex frequency manipulations to further improve the performance, which we leave for future studies. As the first work towards self-supervised learning in the frequency domain, we mainly provide our insights and hope our study can serve as a promising starting point to motivate future work on this direction.
>
> > **Q2:** Although the authors attempt to provide a unified perspective of the method by relating to denoising, super-resolution and deblurring (Section 3.2), such perspective is very natural to have since most image processing methods in the spatial domain would have a corresponding interpretation in the frequency domain.
>
> **A2:** We would like to clarify that the correspondence between the spatial domain and the frequency domain is exactly what our motivation lies in. Such a correspondence motivates us to directly perform masking in the frequency domain, which potentially captures a wider range of information that is covered by multiple corruptions and leads to better performance than these spatial-domain-based image processing tasks. Although it is natural and intuitive to have such a perspective, it has never been explored before. We are the first to study it, and further introduce a flexible and unified MFM method that is applicable to various architectures.
>
> > **Q3:** The authors claim that no extra model (e.g., momentum teacher) and no mask token are advantages of MFM over other self-supervised methods. However, these differences are more about implementation choices rather than principled advantages.
>
> **A3:** We would like to clarify that, the mentioned points we claim are to demonstrate that, without relying on mask tokens or more complex designs (e.g., momentum encoder, discrete visual tokens, etc.) like previous methods do, a simple mask-frequency strategy can also achieve competitive performance for both ViT and CNN architectures. The mentioned design choices make existing self-supervised learning (SSL) methods either computationally expensive (due to using extra model) or not applicable to broader architectures like CNNs (due to using mask tokens). Without using these design choices makes MFM a quite flexible and unified pretext task for various architectures, which are also our principled advantages. We would like to provide a different perspective to existing SSL works and hope our work can motivate the community to rethink the necessity of using more complex design choices.
>
> > **Q4:** Why is no gain observed for CNNs?
>
> **A4:** As analyzed in the “ResNet-50” paragraph in Section 4.4.1 (page 8), the degraded performance of SR, deblurring and denoising is due to the architectural difference between ViT and CNN. Compared with ViT, the convolution operation in CNN tends to be more effective in capturing high-frequency components. Thus, encouraging a CNN model to reconstruct high-frequency components of images brings no benefits to the performance. In contrast, our MFM can outperform its supervised counterparts in both ViT and CNN architectures as it leverages both low- and high-frequency components.
>
> > **Q5:** Why could masked frequency modeling achieve better robustness than MAE?
>
> **A5:** Many prior works (e.g., [1, 2, 3]) have shown that frequency is largely correlated with improving model robustness. Thus, compared with the spatial masking, manipulating in the frequency domain can be a more direct and promising way to improve robustness.
>
> [1] A Fourier Perspective on Model Robustness in Computer Vision. In NeurIPS, 2019.
>
> [2] Improving Robustness against Common Corruptions with Frequency Biased Models. In ICCV, 2021.
>
> [3] Fourier-Based Augmentations for Improved Robustness and Uncertainty Calibration. In NeurIPS, 2021.

---

> ### Author Response · Authors · 2022-12-01
> **Looking forward to your feedback**
>
> Dear Reviewer 34gh,
>
> We sincerely thank you for the review and comments. We have posted our response to your initial comments, which we believe has covered your concerns. We are looking forward to your feedback on whether our answers have addressed your concerns or you have further questions.
>
> Thank you!
>
> Authors

---

### Official Review · Reviewer_Un1w · 2022-10-24

**Confidence:** 4
**Correctness:** 3
**Technical Novelty And Significance:** 3
**Empirical Novelty And Significance:** 2
**Recommendation:** 5

**Clarity, Quality, Novelty And Reproducibility:**

Paper is clear an easy to follow. The proposed empirical study appears novel to me, and authors say they will release code and pretrained models to foster reproducible research.

On the quality side, few ablations could be added to better demonstrate the value of the approach:
- It would be nice to add a direct comparison with random patches dropping in table 1 to directly compare MFM to patches-masked denoising.
- It is surprising that MFM only uses a linear decoder as other auto-encoders to use a deep decoder. It would be also nice to explore the impact of the decoder depth in the ablation.

I have additional questions:
- Why is the loss computed in the FFT domain as it should be somewhat similar to compute it in RGB space. Did you try the later? Do you expect a certain inductive bias by working in the frequency domain.
- Do you need to convert back the image to RGB domain for pretraining? Couldn’t you apply the FFT transform to the input for the finetuning stage?


**Strength And Weaknesses:**

Strengths:
- To my knowledge, the proposed masking strategy is novel and sound.
- Authors evaluate their representation on various downstream tasks requiring different level of abstraction (classification, segmentation).
- Authors propose an ablation of the masking design choice.

Weaknesses:
- One of the main advantage of the masked auto-encoder is its scalability as the encoder does not need to process the masked patches. The current MFM proposal requires to process the full-images and therefore comes with higher computational cost. It is not clear to me if this approach is viable to train larger model such as VIT.L or VIT.H.
- Some baselines are missing in the experimental study. Data2Vec (Baevski et al., 2022) for instance, is a masked image modeling approach which achieves 84.2 top-1 on imagenet finetuning with a VIT.B. MSN (Assran et al., 2022) is another approach combining mask image modeling and siamese networks which exhibits good robustness results (see table 14 in their paper).
Given that those approaches using patches masking obtain good results, it is unclear to me why one should prefer the use of mask frequency modeling.



**Summary Of The Paper:**

This paper proposes an empirical study of different masking-noise strategies for an image auto-encoder model. In contrast to the recent masked auto-encoded, authors propose to mask the inputs in the frequency domain by randomly dropping high or low frequency components. Their approach learns a representation in a self-supervised manner by reconstructing the missing component in the frequency domain.

Authors demonstrates the soundness of their approach on ImageNet finetuning, ADE20K segmentation task and various robustness tasks where they show comparable or better results compared to the previous work.


**Summary Of The Review:**

The paper proposes an empirical study of the masking noise in auto-encoder which seems novel and is an interesting study. However, the practical impact of the study is somewhat limited. It is unclear to me why one practitioner should prefer masking in the frequency domain compared to the patches masking scheme, as the later provides good scalability or can leads to strong results as Data2Vec or MSN demonstrate.

=== After reading rebuttal.

Thank you for your rebuttal. While I appreciate the additional ablation experiments that compare MFM with MIM and different predictor depths, my main concern regarding the paper is not fully addressed.  It is unclear to me why one practitioner should prefer masking in the frequency domain compared to other patch masking scheme,

---

> ### Author Response · Authors · 2022-11-18
> **Response to Reviewer Un1w (Part 2/2)**
>
> > **Q4:** It would be also nice to explore the impact of the decoder depth in the ablation.
>
> **A4:** Thank you for the suggestion. We provide the ablation of decoder depth below (“block(s)” denotes the number of Transformer blocks with a hidden size of 384 and 12 heads):
>
> | Decoder | Top-1 acc (%) 	|
> |----------|:-------------:|
> | linear   	|      **83.1**     	|
> | 1 block 	|      83.0     	|
> | 2 blocks 	|      83.1      	|
> | 4 blocks 	|      83.1     	|
> | 8 blocks 	|      83.1     	|
>
> A simple linear layer for our approach performs the best with lower training costs. We have added this ablation in Appendix A (page 14) of the revised manuscript.
>
> > **Q5:** Why is the loss computed in the FFT domain as it should be somewhat similar to compute it in RGB space.
>
> **A5:** As shown in Table 1(f), we have tried the spatial loss but it performs worse than the frequency loss. Since we perform masking in the frequency domain, directly predicting the missing frequencies in the frequency domain better aligns to our task.
>
> > **Q6:** Do you need to convert back the image to RGB domain for pre-training? Couldn’t you apply the FFT transform to the input for the fine-tuning stage?
>
> **A6:** As stated in footnote 2 (page 4) of the revised manuscript, after performing masking in the frequency domain, we convert back the image to the RGB domain for pre-training. Thus, we do not need to apply the FFT transform to the input during the fine-tuning stage.

---

> ### Author Response · Authors · 2022-11-18
> **Response to Reviewer Un1w (Part 1/2)**
>
> Thank you for the constructive comments. Please find the following for our response.
>
> > **Q1:** MFM requires to process the full images and therefore comes with higher computational cost. It is not clear to me if this approach is viable to train larger model such as ViT-L or ViT-H.
>
> **A1:** We would like to clarify that although MFM takes the full-image as input, it is still computationally efficient as it only uses a very lightweight linear layer as prediction head compared with MAE that uses a much heavier decoder (i.e., 8 Transformer blocks with a hidden size of 512 and 16 heads). More importantly, taking only visible patches as input breaks the regular 2D structure of images, which makes the MAE-style design not applicable to broader architectures like CNNs. In contrast, the main advantage of MFM is that it is agnostic to architectures and can be flexibly applied to both ViT and CNN families. Here, as suggested, we further use ViT-L/16 as an example to show that MFM is viable to train larger models as well. The results are listed below:
>
> | Method  | Epochs 	| Top-1 acc (%) |
> |--------- |:------: |:-------------: |
> | Scratch 	|    -    	| 82.6|
> | MoCo v3 |   600  	| 84.1|
> | MAE     	|   300  	| 84.3|
> | MFM     	|   300  	|**84.5**|
>
> All unsupervised models are based on 300-epoch pre-training on ImageNet-1K for a fair comparison (the effective number of pre-training epochs for MoCo v3 is 600 as it processes two views at a time). MFM still achieves competitive performance compared with the contrastive-based MoCo v3 that is more computationally expensive and the MIM-based MAE that is only limited to the ViT architecture.
>
> > **Q2:** Some baselines are missing in the experimental study.
>
> **A2:** We wish to highlight that it is **unfair** to compare the mentioned methods to ours since they are hybrid methods that combine existing MIM with contrastive learning or clustering into a Siamese network even plus multi-crop augmentation. These more complex technical designs can definitely improve the vanilla MIM performance significantly.
>
> For instance, data2vec (Baevski et al., 2022) additionally uses losses of multiple intermediate Transformer layers, without which its top-1 fine-tuning accuracy is 83.0% (vs. 83.1% achieved by MFM). MSN (Assran et al., 2022) also uses stronger data augmentations including MultiCrop (11 views in total), ColorJitter and GaussianBlur to improve the robustness, without which it will have a shortcut solution. Therefore, to eliminate the interference of other design factors, we mainly compare with pure MIM methods in our paper. More advanced techniques used in these works may also be incorporated into MFM to further improve the performance, which is, however, beyond the focus of this work. Moreover, data2vec and MSN are also limited to the ViT architecture and cannot be applied to CNNs, whereas MFM is a more flexible and unified approach applicable to various architectures. We have added the discussion with these hybrid methods in Appendix D (page 15) of the revised manuscript.
>
> More importantly, we would like to clarify that the aim of this paper is **not** to beat the rapidly changed state-of-the-art performance but provide rather different insights that other low-level tasks beyond MIM can also work well. Compared with a large body of recent works that focus on improving existing MIM approaches by combining with other existing paradigms (e.g., contrastive learning, clustering), we hope our work can provide a different perspective and serve as a promising starting point for exploring flexible and unified visual representation learning of various architectures.
>
> > **Q3:** It would be nice to add a direct comparison with random patches dropping in Table 1 to directly compare MFM to patch-masked denoising.
>
> **A3:** Thank you for the suggestion. We replace the frequency masking with the spatial masking for a direct comparison and the results of ImageNet-1K top-1 fine-tuning accuracy are listed below:
>
> | Task |   Arch.   | Top-1 acc (%) 	|
> |------|---------|:-------------:|
> | MIM  	|  ViT-B/16 	|      82.8     	|
> | MFM  	|  ViT-B/16 	|      **83.1**     	|
> | MIM  	| ResNet-50 	|      77.7     	|
> | MFM  	| ResNet-50 	|      **78.5**     	|
>
> Applying MIM with spatial masking to ResNet-50 performs even worse than the supervised baseline (i.e., 78.1% in RSB A3). In contrast, MFM outperforms the supervised baseline as well as the MIM counterpart regardless of architectures, demonstrating that frequency masking is indeed a more flexible and unified option for different architectures. We have added this ablation in Appendix A (page 14) of the revised manuscript.

---

> ### Author Response · Authors · 2022-11-25
> **Further clarification**
>
> Dear Reviewer Un1w,
>
> Thank you for the feedback. Regarding your concern, we would like to emphasize that MFM is **a more flexible and unified approach applicable to both ViT and CNNs compared with MIM that is only limited to the ViT architecture**. This is the reason why “one practitioner should prefer masking in the frequency domain compared to other patch masking scheme”. More importantly, we believe our work provides a new perspective to existing works and brings different knowledge and values to the community, which is more important than chasing state-of-the-art numbers. We would like to refer the reviewer to another work CIM (https://openreview.net/forum?id=09hVcSDkea), which also explores other possibilities beyond MIM and does not achieve state-of-the-art performance. However, this does not influence its novelty and the new knowledge added to the community as acknowledged by its reviewers. The same rule applies to our work. Moreover, compared with CIM that uses a trainable generator to corrupt the input images, which adds nontrivial pre-training overhead, MFM can achieve comparable performance with CIM while being much more efficient. This further demonstrates the advantage of MFM. For the first time, we demonstrate that a simple non-Siamese method can achieve competitive performance on both ViT and CNNs without more complex designs. We hope the reviewer could reconsider the values of our work.

---

### Official Review · Reviewer_Lxfd · 2022-11-02

**Confidence:** 4
**Correctness:** 3
**Technical Novelty And Significance:** 3
**Empirical Novelty And Significance:** 4
**Recommendation:** 8

**Clarity, Quality, Novelty And Reproducibility:**

Clarity: As a said above, the paper is clearly written, with clean illustrations and nice ablations.

Novelty: It is definitely a new and neat exploration to masked frequency modeling. I haven't seen any work sufficiently similar to this.

Reproducibly: The paper promises to release the code and pre-trained models. It also provides hyper-parameters in sufficient details. So I don't see a problem for reproducibility here.

**Strength And Weaknesses:**

Strengths:
- The exploration of a simple masked frequency modeling algorithm is interesting and valuable to the community, given the recent success of masked image modeling methods for representation learning.
- The paper's exploration is quite complete, covering a simple basic method for MFM, MFM in the context of previous methods, different architectures, different tasks and benchmarks.
- The paper is also relatively well-written and well-polished.

Weaknesses:
- In an ideal world, the newly proposed approach should advance the state-of-the-art in a meaningful way. Unfortunately, this work is still yet to reach that level of performance.
- It would also be great if bigger models are trained to show the power of pre-training (if the computation permits).
- With MAE-style design, MIM is made very efficient that it does not need to compute for all the patches in the encoder. Right now MFM still looks at the full-sequence. This is a potential downside in speed.
- (minor) At page 6 last paragraph, the paper states an hypothesis about the mask shape being largely correlated with the category statistics of pre-training datasets. Is there evidence for this?

**Summary Of The Paper:**

The paper investigates the possibility of masked frequency modeling for representation learning. It covers several research topics centered on MFM: a method that does MFM; connecting previous pre-training methods to MFM (e.g., super resolution); architecture change (from ViT to ConvNets) as an advantage of MFM over masked image modeling (the more popular methods for self-supervised pre-training); more tasks and benchmarks beyond classification with MFM pre-training. The results are not state-of-the-art per-se, but the complete assessment of MFM is interesting and of value.

**Summary Of The Review:**

Overall I think this paper has already reached its maturity. It has a simple, basic method for MFM, and has explored a lot centered around this simple methods (I especially appreciate the connection to other pre-training methods, and experiments on ResNet). Although the method has not advanced state-of-the-art, I would still vote for acceptance given the added knowledge and value from this work to the research community.

---

> ### Author Response · Authors · 2022-11-18
> **Response to Reviewer Lxfd**
>
> Thank you for the constructive comments. Please find the following for our response.
>
> > **Q1:** In an ideal world, the newly proposed approach should advance the state-of-the-art in a meaningful way. Unfortunately, this work is still yet to reach that level of performance.
>
> **A1:** We would like to clarify that the aim of this paper is **not** to beat the rapidly changed state-of-the-art performance but provide rather different insights that other low-level tasks beyond MIM can also work well. Compared with a large body of recent works that focus on improving existing MIM approaches, we hope our work can provide a different perspective to draw the community’s attention beyond MIM and rethink the role of low-level tasks for unsupervised representation learning.
>
> > **Q2:** It would also be great if bigger models are trained to show the power of pre-training (if the computation permits).
>
> **A2:** We use ViT-L/16 as an example to demonstrate the scaling behavior of MFM as suggested. All unsupervised models are based on 300-epoch pre-training on ImageNet-1K for a fair comparison (the effective number of pre-training epochs for MoCo v3 is 600 as it processes two views at a time). The results are listed below:
>
> | Method  | Epochs 	| Top-1 acc (%) |
> |--------- |:------:|:-------------:|
> | Scratch 	|    -    	|      82.6     	|
> | MoCo v3 |   600  	|      84.1     	|
> | MAE     	|   300  	|      84.3     	|
> | MFM     	|   300  	|      **84.5**     |
>
> MFM still achieves competitive performance compared with the contrastive-based MoCo v3 and the MIM-based MAE.
>
> > **Q3:** With MAE-style design, MIM is made very efficient that it does not need to compute for all the patches in the encoder. Right now MFM still looks at the full-sequence. This is a potential downside in speed.
>
> **A3:** We would like to clarify that although MFM takes the full-image as input, it is still efficient as it only uses a very lightweight linear layer as prediction head compared with MAE that uses a much heavier decoder (i.e., 8 Transformer blocks with a hidden size of 512 and 16 heads). More importantly, taking only visible patches as input breaks the regular 2D structure of images, which makes the MAE-style design not applicable to broader architectures like CNNs. In contrast, the main advantage of MFM is that it is agnostic to architectures and can be flexibly applied to both ViT and CNN families.
>
> > **Q4:** At page 6 last paragraph, the paper states a hypothesis about the mask shape being largely correlated with the category statistics of pre-training datasets. Is there evidence for this?
>
> **A4:** In our preliminary experiments, we observe that different categories tend to have different dominant patterns reflected in the frequency spectrum. For example, for man-made environments (e.g., buildings, cityscapes, etc.), the spectrum tends to have more frequencies in the horizontal and vertical directions since these images mostly contain the horizontal and vertical contents, while the frequency distribution of natural environments (e.g., fauna, flora, etc.) tends to be more uniform. Therefore, in practice, we may further consider using different mask shapes when pre-training on different datasets. This is only our analysis and insight, and we leave more in-depth explorations for future work.

---

> ### Author Response · Authors · 2022-12-01
> **Looking forward to your feedback**
>
> Dear Reviewer Lxfd,
>
> We sincerely thank you for the review and comments. We have posted our response to your initial comments, which we believe has covered your concerns. We are looking forward to your feedback on whether our answers have addressed your concerns or you have further questions.
>
> Thank you!
>
> Authors

---

### Author Response · Authors · 2022-11-18
**The revised manuscript is uploaded**

Dear reviewers,

We sincerely thank you for the valuable reviews and comments. We have uploaded our revised manuscript accordingly to address your concerns. We highlighted the revised contents in red to facilitate the review process. In the following, we will respond to every issue raised by each reviewer. We hope you will find your concerns properly addressed and we welcome any further discussions.

---

### Decision · Program_Chairs · 2023-01-20

**Decision:**

Accept: poster

**Justification For Why Not Higher Score:**

The paper's idea can be interesting and may lead to some rethinking about masked image modelling. But we have to admit that it failed to show advantages in experiments, which makes it unclear why a deep learning practitioner would like to follow this work. Three out of four reviewers thought the paper has its own value to be published, but it would be hard to push an even higher recommendation.

**Justification For Why Not Lower Score:**

The overall idea of exploring masking in the frequency domain can be interesting. Though the current performance of this algorithm is not comparable with those SOTA MAE variants, the paper indeed can provide some inspiration for the following research about masked image modelling for visual representation learning.

**Metareview: Summary, Strengths And Weaknesses:**

This paper studied masked frequency modelling, which was suggested as an alternative way to masked image modelling in the spatial domain. The overall idea is interesting and has been widely recognised by reviewers. But the major concern is its slightly lower performance, compared with those SOTA MAE variants.

**Note From Pc:**

if the above contains the word "oral" or "spotlight" please see: "oral" presentation means -> notable-top-5% and "spotlight" means -> notable-top-25%. As stated in our emails, we are disassociating presentation type from AC recommendations

**Summary Of Ac-Reviewer Meeting:**

Reviewers had a virtual meeting to discuss the strength and weaknesses of this paper. The practical performance advantage over existing works seems to be a major concern shared by reviewers, but reviewers also agreed that the idea of exploring the frequency domain could be inspiring. We took anonymous voting and three out of four reviewers would like to recommend the acceptance of this paper.